# BONGARD-OPENWORLD:
# FEW-SHOT REASONING FOR FREE-FORM VISUAL CONCEPTS IN THE REAL WORLD

**Rujie Wu\*[1]**   **Xiaojian Ma\*[2]**   **Zhenliang Zhang[2]**
**Wei Wang** ✉ [2]   **Qing Li** ✉ [2]   **Song-Chun Zhu[2,3,4]**   **Yizhou Wang[1,4]**
[1]School of Computer Science, Peking University
[2]National Key Laboratory of General Artificial Intelligence, BIGAI
[3]School of Intelligence Science and Technology, Peking University
[4]Institute for Artificial Intelligence, Peking University
**\*** Equal contribution   ✉ Co-corresponding authors
Project page: Bongard-OpenWorld

## ABSTRACT

We introduce **Bongard-OpenWorld**, a new benchmark for evaluating real-world few-shot reasoning for machine vision. It originates from the classical *Bongard Problems (BPs)*: Given two sets of images (positive and negative), the model needs to identify the set that query images belong to by inducing the visual concepts, which is exclusively depicted by images from the positive set. Our benchmark inherits the few-shot concept induction of the original BPs while adding the two novel layers of challenge: 1) open-world free-form concepts, as the visual concepts in Bongard-OpenWorld are unique compositions of terms from an open vocabulary, ranging from object categories to abstract visual attributes and commonsense factual knowledge; 2) real-world images, as opposed to the synthetic diagrams used by many counterparts. In our exploration, Bongard-OpenWorld already imposes a significant challenge to current few-shot reasoning algorithms. We further investigate to which extent the recently introduced Large Language Models (LLMs) and Vision-Language Models (VLMs) can solve our task, by directly probing VLMs, and combining VLMs and LLMs in an interactive reasoning scheme. We even conceived a neuro-symbolic reasoning approach that reconciles LLMs & VLMs with logical reasoning to emulate the human problem-solving process for Bongard Problems. However, none of these approaches manage to close the human-machine gap, as the best learner achieves 64% accuracy while human participants easily reach 91%. We hope Bongard-OpenWorld can help us better understand the limitations of current visual intelligence and facilitate future research on visual agents with stronger few-shot visual reasoning capabilities.

## 1 INTRODUCTION

In recent years, substantial progress has been recorded in developing visual intelligence. Given an image, visual agents now can robustly recognize the presenting objects (object detection and segmentation (Deng et al., 2009; He et al., 2017; Lin et al., 2014)), describe what's happening (image captioning (Xu et al., 2015; Li et al., 2023)), and even answering complex questions about it (visual question answering (Goyal et al., 2017; Hudson & Manning, 2019; Ma et al., 2022a)). However, completing many of these tasks requires a massive amount of training data. On the contrary, humans can perform more sophisticated tasks with very few visual inputs (Marr, 2010; Zhu & Zhu, 2021; Huang, 2021). For example, humans can recognize and reason about compositional real-world visual concepts from just a few examples (Lake et al., 2015; Zhang et al., 2019; Nie et al., 2020; Jiang et al., 2022), *i.e.* few-shot visual reasoning. To facilitate human-level visual intelligence, it is necessary to go beyond canonical tasks and develop new benchmarks that aim to comprehensively evaluate few-shot learning of novel and complicated visual concepts.

Several benchmarks have been introduced with a focus on few-shot learning of visual concepts, such as Omniglot (Lake et al., 2015), miniImageNet (Vinyals et al., 2016), and Meta-Dataset (Triantafillou et al., 2020). However, these datasets primarily focus on identifying simple object categories instead

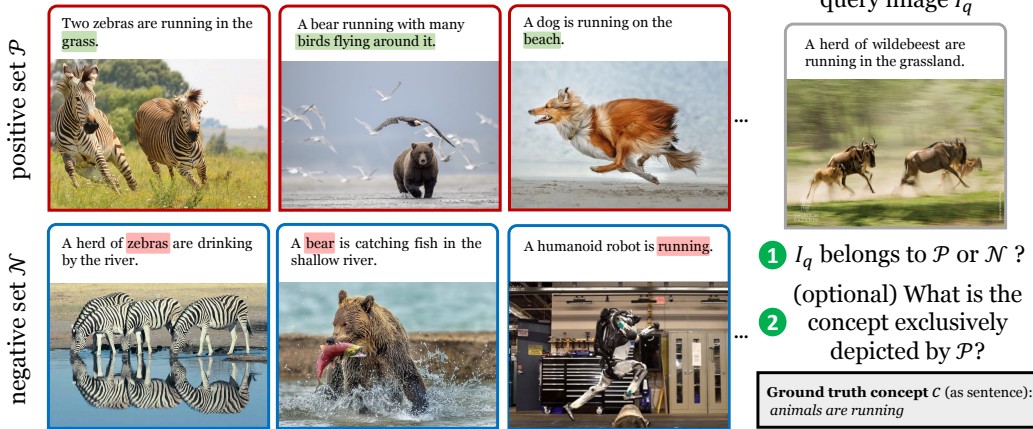

Figure 1: **Task illustration of Bongard-OpenWorld.** Given two set of images $\mathcal{P}$ and $\mathcal{N}$, the model needs to identify which set the query image $I_q$ belongs to by inferring the concepts $\mathcal{C}$ that is exclusively depicted by $\mathcal{P}$. **Note that the captions and the concepts $\mathcal{C}$ won't be provided to the model.** To further increase the difficulty of our task, we introduce *distractors* as additional contents of the positive images other than the concept $\mathcal{C}$, and *hard negatives* to ensure the content of negative images *partially overlaps* with the concepts $\mathcal{C}$. These practices could force the model to reason about the visual concepts by contrasting the positives and the negatives.

of novel and complex visual concepts, *e.g.* compositions of objects and their characteristics (Krishna et al., 2017) and visual relationships (Chao et al., 2015), and generalization of attributes (Ren et al., 2020). Many benchmarks have been dedicated to abstract visual reasoning and are considered to be few-shot learning, such as RPM (Raven-style Progressive Matrices) (Zhang et al., 2019; Barrett et al., 2018) and Bongard Problems (Nie et al., 2020). While offering interesting visual tasks, these benchmarks are prone to use synthetic graphics instead of real-world images and stay with basic object-level geometrical concepts such as shapes, sizes, amounts, etc. The closest benchmark to our work is Bongard-HOI (Jiang et al., 2022), which features few-shot visual reasoning and compositional visual concepts (human-object interactions, *i.e.* HOI) in the real world. But since it is built upon an existing close-vocabulary image recognition dataset (HAKE (Li et al., 2019)), the total number of unique concepts (242, while Bongard-OpenWorld has 1.01k) can be limited. Moreover, the concepts are restricted to be HOIs with a fixed structure ⟨object, interaction⟩, which deviates from the free-form nature of visual concepts in the real world.

To solve the aforementioned issues and step towards a more general few-shot visual reasoning challenge that invites all possible contestants, we introduce **Bongard-OpenWorld**, a benchmark that reconciles the best of all parties: *few-shot learning*, *real-world images*, and *compositional visual concepts*. Bongard-OpenWorld inherits the simplicity of the original Bongard Problems: given six positive and six negative images, along with two query images, the goal is to identify the *visual concepts*, which is exclusively depicted by the positive images, and make binary predictions on the query images (see Figure 1 for illustrations, their quantity here is halved to facilitate a clearer visualization of image details).

At the heart of our benchmark are *open vocabulary free-form concepts* and illustrated in Table 1. Instead of using a small and pre-defined set of concepts that are predominately about the same topic (object attributes in Bongard-LOGO (Nie et al., 2020), HOIs in Bongard-HOI (Jiang et al., 2022), etc.), we leverage Conceptual Captions (CC-3M) (Sharma et al., 2018), a massive web-crawled collection of rich and open vocabulary image descriptions. We further propose *grid sampling* to extract visual concepts from the streamlined captions. The resulting visual concepts are therefore both free-form, with no assumption on the structure, *e.g.* 2-tuple as in Bongard-HOI, and open vocabulary. We also crowd-source some challenging concepts, including abstract visual attributes, and commonsense factual knowledge (examples can be found in Table 1), and compose them with the concepts extracted from CC-3M. Finally, we manually verify the plausibility of our generated visual concepts. We end up with 1.01K unique concepts, and 26.6% of them are crowd-sourced challenging concepts. Statistics of these visual concepts can be found in Figure 2.

We then construct Bongard-OpenWorld problems out of the aforementioned visual concepts. Each problem is assigned a positive concept and an online image search tool is used to find the most relevant images. Additionally, we follow the practice in (Jiang et al., 2022) to introduce *distractors* and *hard negatives*. We collect positive images containing additional content as distractions other than the visual concepts $\mathcal{C}$. Moreover, by partially modifying the positive concept to produce negative concepts, we ensure the content of the negative images partially overlaps with the positives. Therefore,

Table 1: **A catalog of visual concepts in Bongard-OpenWorld**. We demonstrate the categories of concepts mined from CC-3M, and challenging commonsense-related concepts that are crowd-sourced and augmented to the dataset. *denotes commonsense. More examples see Table 9.

| Concept Category | ID | Example | Concept Category | ID | Example |
|---|---|---|---|---|---|
| Anything else* | 0 | Animals are running. | And / Or / Not | 5 | A man without beard. |
| HOI | 1 | A person playing the guitar. | Factual Knowledge | 6 | A building in US capital. |
| Taste / Nutrition / Food | 2 | A plate of high-calorie food. | Meta Class | 7 | Felidae animals. |
| Color / Material / Shape | 3 | A wooden floor in the living room. | Relationship | 8 | A bench near trees. |
| Functionality / Status / Affordance | 4 | An animal capable of flying in the tree. | Unusual Observations | 9 | Refraction of light on a glass cup. |

reasoning about the visual concepts by contrasting the positive and negative images will be required to solve the problem, rather than merely recognizing some simple common contents among the positive images (illustrated in Figure 1). Overall, we produce 1.01K high-quality Bongard-OpenWorld problems. Thanks to the diverse set of concepts, each problem has its unique visual concepts. Comparisons with counterpart benchmarks can be found in Table 2.

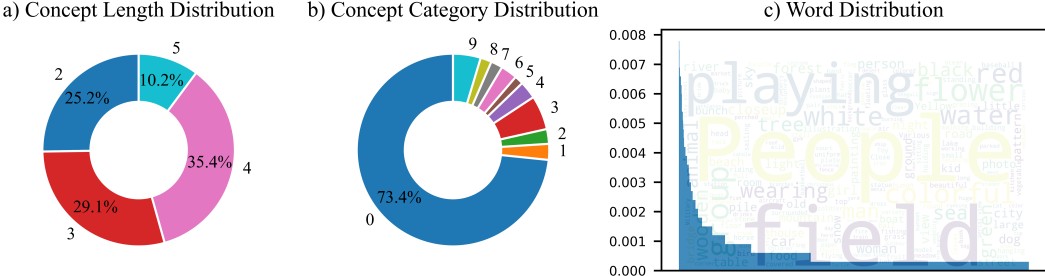

Figure 2: **Statistics of Bongard-OpenWorld.** Our benchmark exhibits a range of concept lengths, spanning from 2 to 5 (as depicted in subfigure a), with an average length of 3.3. As demonstrated in Table 1, crowd-sourced commonsense concepts take ID 1~9, with 0 indicating "anything else" (as depicted in subfigure b). While some words are more frequent (see the word cloud, as depicted in subfigure c), the overall frequency of words in Bongard-OpenWorld concepts follows a long-tailed distribution.

**Benchmarking existing few-shot reasoners**: In our experiments, we examine state-of-the-art few-shot learning approaches, including non-episodic, meta-learning, and transformer-based models. Although the best few-shot learner (SNAIL (Mishra et al., 2018)) shows some promising results on Bongard-OpenWorld with a 64% average accuracy (the chance performance is 50%), the overall gap to human performances (91% of human contestants) is still substantial. To understand the failure mode, we hypothesize that visual pretraining could be pivotal due to the open vocabulary nature of our task. Therefore we investigate combining the learner with several pretrained visual representations. The results confirm that few-shot learners fueled with proper open-ended pretrained models, *e.g.* CLIP (Radford et al., 2021) can alleviate this gap. Further, we investigate to which extent the recently introduced Vision-Language Models (VLMs) and Large Language Models (LLMs) can approach Bongard-OpenWorld, by directly probing VLMs, and combining VLMs and LLMs in an interactive reasoning scheme (Zhu et al., 2023; Gao et al., 2022). We also developed a neuro-symbolic reasoning method that integrates LLMs and VLMs with logical reasoning, it aims to mimic human problem-solving processes in addressing Bongard Problems. Our results indicate that despite the impressive results these approaches have attained in other tasks, they can still be confused by similar content between positive and negative examples in Bongard-OpenWorld and therefore fail to close the human-machine gap.

Our contributions are summarized as follows:

- We introduce Bongard-OpenWorld, a new benchmark for few-shot visual reasoning with visual concepts in the real world, aiming at reconciling the challenging capabilities of few-shot learning and reasoning with abstract and complicated real-world visual concepts and facilitating the development of human-like visual intelligence.

- We carefully curate Bongard-OpenWorld to include open vocabulary and free-form visual concepts, ranging from object categories to abstract visual attributes and commonsense factual knowledge. We also use distractors and hard negatives to make merely recognizing some common contents in the positives insufficient to complete our tasks.

- We conduct extensive analysis on state-of-the-art few-shot reasoners including canonical few-shot learning systems and powerful LLMs, VLMs, and even neuro-symbolic learners. However, empirical results indicate that these approaches are struggling to match human performances on Bongard-OpenWorld. Our findings suggest that robustly capturing sophisticated (compositional, abstract, factual or commonsense, *etc.*) visual concepts across multiple visual stimuli can still be a huge challenge to even today's vision models.

Table 2: **An overview of benchmarks covering free-form image concepts, few-shot learning, and visual reasoning**. The abbreviation *vocab.* denotes *vocabulary* and *attr.* denotes *attributes*. *We consider a benchmark to be open vocabulary when it is collected without assuming a fixed set of visual concepts (*e.g.* CC-3M (Sharma et al., 2018)), or the assumed set is substantially large (*e.g.* Meta-Dataset (Triantafillou et al., 2020)). **The object attributes and object counts can be freely composed in V-PROM (Teney et al., 2020) but the object (O) and interaction (I) in Bongard-HOI (Jiang et al., 2022) cannot.

| dataset | concept | free-form concept | open vocab.* | real-world images | few-shot | hard negatives | #concepts | #tasks |
|---------|---------|-------------------|--------------|-------------------|----------|----------------|-----------|--------|
| CC-3M (Sharma et al., 2018) | image caption | ✓ | ✓ | ✓ | ✗ | ✗ | 31.1K | 3.3M |
| Omniglot (Lake et al., 2015) | shape | ✗ | ✗ | ✗ | ✓ | ✗ | 50 | 1.62K |
| miniImageNet (Vinyals et al., 2016) | image label | ✗ | ✗ | ✓ | ✓ | ✗ | 100 | 60K |
| Meta-Dataset (Triantafillou et al., 2020) | image label | ✗ | ✓ | ✓ | ✓ | ✗ | 4,934 | 52.8M |
| RPM (Barrett et al., 2018) | shape | ✗ | ✗ | ✗ | ✓ | ✗ | 50 | 11.36M |
| Bongard-LOGO (Nie et al., 2020) | shape | ✗ | ✗ | ✗ | ✓ | ✗ | 627 | 12K |
| V-PROM (Teney et al., 2020) | attr. & count | ✓** | ✗ | ✓ | ✓ | ✗ | 478 | 235K |
| Bongard-HOI (Jiang et al., 2022) | HOI | ✗** | ✗ | ✓ | ✓ | ✓ | 242 | 53K |
| **Bongard-OpenWorld (ours)** | image caption | ✓ | ✓ | ✓ | ✓ | ✓ | 1.01K | 1.01K |

## 2 THE BONGARD-OPENWORLD BENCHMARK

A problem instance in Bongard-OpenWorld can be formulated as a tuple $\langle \mathcal{C}, \mathcal{P}, \mathcal{N}, I_q \rangle$, where $\mathcal{C}$ denotes the free-form compositional visual concepts, *e.g. animals are running* in Figure 1. The $\mathcal{C}$ is exclusively depicted by the positive images $\mathcal{P}$, while all images from the negative set $\mathcal{N}$ do not depict it but might partially contain some of its terms $c$. The task is to make a binary prediction on the query image $I_q$: to determine whether $\mathcal{C}$ can be found in $I_q$ or not. While it is optional, the learner could explicitly induce the visual concepts $\mathcal{C}$ via captioning to offer some explainability of its prediction on $I_q$. The following sections will detail how to collect the visual concepts and build the benchmark.

### 2.1 COLLECTING FREE-FORM OPEN VOCABULARY VISUAL CONCEPTS

**Visual concepts $\mathcal{C}$ in Bongard-OpenWorld.** We begin with a more detailed illustration of the visual concepts $\mathcal{C}$. As we anticipate it to be *free-form* and *open vocabulary* in our benchmark, $\mathcal{C}$ is effectively an arbitrary *sentence* that describes the content depicted by all images from the positive set $\mathcal{P}$ exclusively. However, since not all words in the sentence are meaningful to its *semantics*, we may streamline and convert it into a *tuple of words* and define the *length of visual concepts $\mathcal{C}$* as the number of words in the tuple. Longer $\mathcal{C}$ will be more difficult to be identified.

**Grid-sampling of visual concepts from CC-3M.** We propose to extract these visual concepts from CC-3M (Sharma et al., 2018), a massive dataset of image-text pairs with a comprehensive collection of open vocabulary free-form image descriptions. We then streamline all the captions into concepts tuples and perform *grid sampling*. Due to space limitations, we only provide its key operations here and more details and pseudo-code can be found in the Appendix C: 1) We run sliding window with size 2/3/4/5 over all the concepts tuples to count the frequency of concepts with length 2/3/4/5; 2) Instead of sampling from the whole CC-3M "concept tuples" (refer to candidates for the visual concepts, where each concept conprises a tuple of words, ex. <red, tie>), we first split it into seveal small pools, or as we name it, "grids" (denotes the pool we sample concept from), with size of 300 each. Then we perform top-k sampling within each grid to obtain concepts (this process is termed *grid sampling*). We find this balances the need for sampling top concepts and sample diversity, as the variance introduced by a small grid facilitates the sampling of more long-tailed concepts in CC-3M.

**Crowd-sourcing for challenging visual concepts.** During our early inspection, we observed that visual concepts out of CC-3M are mostly about object categories and their relatively simple attributes & relations. However, humans can understand more abstract concepts that require *commonsense factual knowledge*. Therefore, we further augment the visual concepts with these challenging concepts through crowd-sourcing. Specifically, the annotators are instructed to write visual concepts by following a predefined set of categories illustrated in Table 1. They are also asked to combine these challenging concepts with those mined from CC-3M. Our experiments confirm that Bongard-OpenWorld problems with these commonsense concepts become more difficult to solve.

### 2.2 FROM VISUAL CONCEPTS TO REAL-WORLD BONGARD PROBLEMS

**Distractors in positives and hard negatives.** To further increase the intra-diversity among the positives $\mathcal{P}$ and therefore perplex the induction of given visual concepts $\mathcal{C}$, we prompt ChatGPT to expand it into 10 *sentences for positives* by inserting *distracting* objects, attributes, etc. while ensuring common ground is still $\mathcal{C}$. Moreover, prior work on Bongard Problems (Jiang et al., 2022)

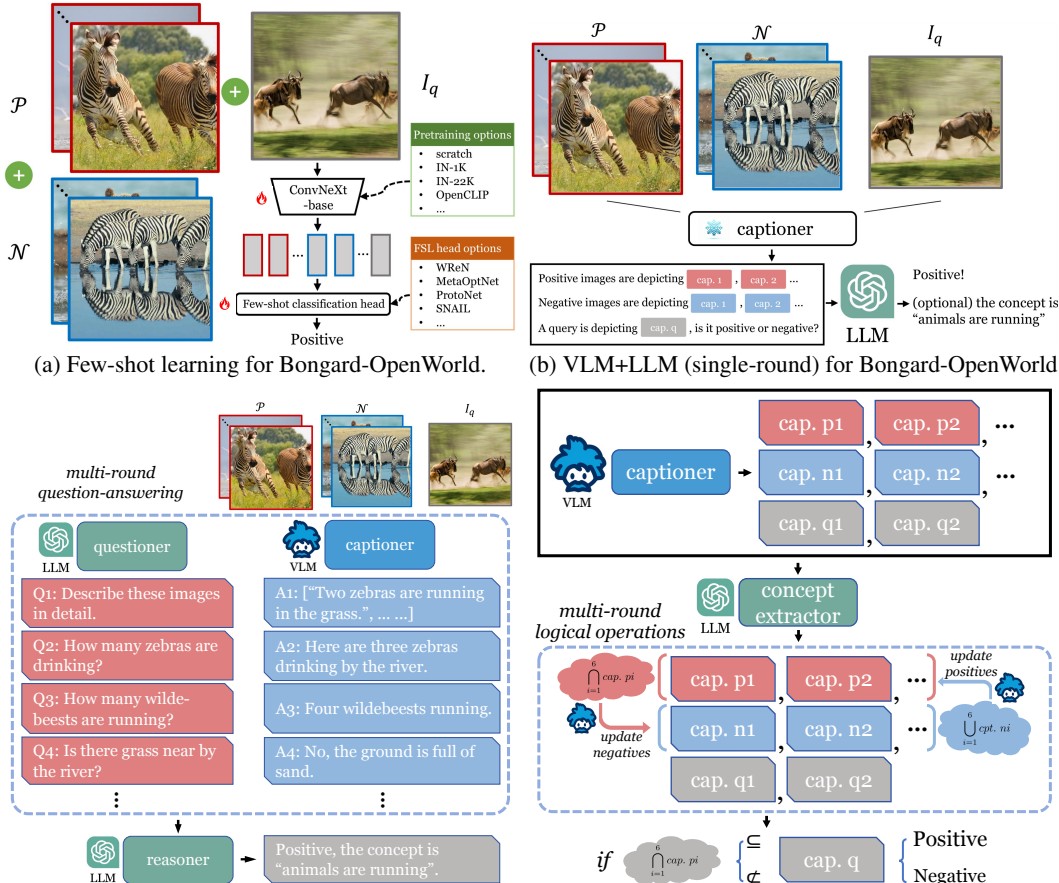

(a) Few-shot learning for Bongard-OpenWorld.    (b) VLM+LLM (single-round) for Bongard-OpenWorld

(c) VLM+LLM (multi-round) for Bongard-OpenWorld  (d) Neuro-symbolic approach for Bongard-OpenWorld

Figure 3: **Models for Bongard-OpenWorld.** We explore four families of approaches: (a) casting Bongard-OpenWorld into a standard "2-way, 6-shot" few-shot learning problem and tackling it using state-of-the-art few-shot learners with pretrained image representations; (b) combining an LLM (reasoner) and a VLM (image captioner) in a single round fashion, where the VLM simply caption each Bongard image and send their captions to LLM for solving this problem; (c) extending the method in (b) to multiple rounds, where the LLM will also iteratively probe the VLM for more image details, resulting in more condense information for solving Bongard; (d) neuro-symbolic approach, where a VLM generates the initial captions, then an LLM extracts visual concepts from them. These concepts are subsequently updated through logical operations, leveraging the responses provided by VLM, until the problem is solved. Zoom in for a better view.

has shown that improper choices of the negative set $\mathcal{N}$, *e.g.* a set of randomly chosen images, could trivialize the challenge by allowing the model to merely recognize part of the visual concepts $\mathcal{C}$ (*e.g.* only recognizing "animals" of a full concept "animals are running" in Figure 1) and solve the problem. Therefore, the content of the negative images should overlap with $\mathcal{C}$ to ensure the need of inducing full visual concepts. To this end, we again prompt ChatGPT to edit $\mathcal{C}$ into 10 *sentences for negatives* that only *partially overlap* with it. Both sets will then be used to collect images.

**Image collection and adversarial query selection.** We ask our annotators to pick some sentences from the two sets, use an online image search tool to find relevant images, and construct the positive and negative set of a Bongard-OpenWorld problem by following the principle (the visual concepts $\mathcal{C}$ are exclusively depicted by the positives $\mathcal{P}$). The annotators are then asked to provide two sets of candidates for positive and negative queries based on the unused positive and negative sentences. Finally, we compute the embedding of all images using CLIP (Radford et al., 2021) and the image with maximal embedding distance to the mean of positives will be selected as a positive query (similar to negative query). We find this adversarial query selection help ensure the difficulty of the problem.

**Final curation.** We employ a round of manual refinement as the last hurdle to help us curate the dataset. We ask the participant to flag any problem that does not comply with the Bongard principle and provide a fix, by replacing the faulty images or modifying the visual concepts $\mathcal{C}$ if possible.

### 2.3 DATA STATISTICS, AND METRICS

**Statistics.** We present the statistics of Bongard-OpenWorld in Figure 2. Here are some observations: First of all, although our dataset does not assume the structure of the free-form visual concepts $\mathcal{C}$, concepts with lengths 2, 3, and 4 accounts for a staggering 90% of all the problems, which both reflect the nature of common visual concepts and avoid overly complicated problems when the concepts become longer. The even distribution of concepts these lengths ensures a reasonable difficulty and is verified by our human study in Section 4.2; Moreover, we manage to crowd-source a significant amount of interesting commonsense visual concepts, which make up nearly 1/4 of all the tasks, and each proposed commonsense category has a fair share in the problems; Finally, the word cloud and histogram of concepts demonstrate that some words, *e.g.* "people", "field", etc., can be more popular but the word distribution is long-tailed. This aligns with CC-3M, *i.e.* lots of human-centered scenes and the concepts indeed follow a long-tailed distribution.

**Dataset splits and metrics.** We divide the 1.01k Bongard-OpenWorld problems into a 610/200/200 split for training/validation/test. We use rejection sampling to ensure the distribution of concept lengths, commonsense vs. non-commonsense concepts are identical among these three sets. Following the prior work in Bongard Problems (Jiang et al., 2022; Nie et al., 2020), we report the overall accuracy of all models. To help understand the difficulty imposed by the free-form open vocabulary concepts, we further report the accuracy on some subsets: 1) problems with short concepts, *i.e.* concept length less or equal to three; 2) problems with long concepts, *i.e.* concept length more than three; 3) problems with commonsense concepts; 4) problems with non-commonsense concepts. These additional metrics provide a more detailed and nuanced understanding of the models' performance and can help identify specific strengths and weaknesses of each model. We also introduce a set of metrics that measure the correctness of the visual concepts explicitly induced by the model. Please refer to the Appendix C for detailed descriptions of these metrics.

## 3 MODELS FOR BONGARD-OPENWORLD

In general, Bongard-OpenWorld can be viewed as a few-shot learning problem. Given two small sets of images $\mathcal{P}$ and $\mathcal{N}$, the model ultimately needs to make a binary prediction on the query image $I_q$. A representation network is also needed to process the images and provide input to the few-shot learner. We start by considering several canonical few-shot learners, including non-episodic approaches and meta-learning-based avenues. Inspired by the recent progress in Large Language Models (LLMs) and Vision-Language Models (VLMs), we investigate to which extent they can approach Bongard-OpenWorld, by directly probing VLMs, and combining VLMs and LLMs in an interactive reasoning scheme (Zhu et al., 2023; Gao et al., 2022). We also designed a neuro-symbolic reasoning approach that reconciles VLMs and LLMs with logical reasoning. We provide an overview of these methods in Figure 3.

### 3.1 FEW-SHOT LEARNING FOR BONGARD-OPENWORLD

**Few-shot learning methods.** Following the seminal few-shot learning work (Nie et al., 2020; Jiang et al., 2022; Triantafillou et al., 2020; Requeima et al., 2019; Bateni et al., 2020), we consider *meta-learning method*, where the learner adopts the episodic learning setting and learns to train a classifier using the support set with a *meta objective* and evaluate the trained classifier on the query. In our experiment, we include the following meta learners: 1) *ProtoNet* (Snell et al., 2017) and *Meta-Baseline* (Chen et al., 2020), which both feature a metric-based meta objective; 2) *MetaOptNet* (Lee et al., 2019) and *SNAIL* (Mishra et al., 2018), two optimization and memory-based methods. Note that *SNAIL* uses transformers (Vaswani et al., 2017) and delivers strong results in many few-shot learning tasks and outperforms other baselines on Bongard-OpenWorld. Readers are encouraged to refer to the Appendix F and the relevant papers for more details.

**Auxiliary task of captioning.** As we mentioned before, understanding visual concepts serves as the foundation of completing Bongard-OpenWorld tasks. Therefore we investigate if we can boost this via an auxiliary task of predicting the concepts. Since we employ free-form open vocabulary concepts, we connect our image encoder to a pretrained BLIP-2 (Li et al., 2023) caption decoder (specifically, the QFormer and the language model) and use caption loss for the task. The overall loss then becomes $\mathcal{L} = \mathcal{L}_{\text{Bongard}} + \alpha\mathcal{L}_{\text{caption}}$, where $\mathcal{L}_{\text{Bongard}}$ and $\mathcal{L}_{\text{caption}}$ denote the main loss and the auxiliary loss, respectively. $\alpha$ is a trading-off weight. More details can be found in the Appendix D.

## 3.2 COMBINING VLMS AND LLMS FOR BONGARD-OPENWORLD (SINGLE-ROUND)

**Evaluation.** In this approach, an LLM is used as a few-shot reasoner while a VLM is invoked to translate the images in a Bongard Problem into captions. Specifically, we evaluate two publicly available LLMs: ChatGPT (Ouyang et al., 2022) and GPT-4 (Bubeck et al., 2023), and follow the prior practices (Ma et al., 2022b; Gao et al., 2022) to use BLIP-2 (Li et al., 2023) and its successor, InstructBLIP (Dai et al., 2023) to extract captions. Finally, the LLM is anticipated to produce a binary prediction of the query image by learning from the few-shot examples (images as captions) in the prompt through in-context learning (Brown et al., 2020). We provide more details on the prompt design in the Appendix D.

**Explicit visual concepts induction.** Since we are particularly interested in the visual concepts induced by the models, we further prompt the LLMs to produce an explanation of the binary prediction it makes. Specifically, the model is guided to summarize the visual concepts $\mathcal{C}$ that is exclusively depicted by the positives. The model's summary will then be compared against the ground truth and evaluated using the image captioning metrics we introduced in Section 2.3.

## 3.3 COMBINING VLMS AND LLMS FOR BONGARD-OPENWORLD (MULTI-ROUND).

In the previous section, LLM indeed relies on the captions produced by the VLM. However, these captions may lack crucial image information, potentially misleading the LLM reasoner. To mitigate this issue, an additional task can be assigned to the LLM beforehand. This task involves generating questions based on the initial captions and the objectives of BPs. These questions are not hallucinations from the LLMs; they are derived from existing information. The VLM utilizes these questions as prompts, providing answers based on the image content, and feedback to the LLM to update existing information. By iteratively repeating this question-answering process multi-round, the LLM can acquire highly detailed information that enhances their reasoning. In our results, we denote this approach as ChatCaptioner, as it is introduced in Zhu et al. (2023).

## 3.4 A NEURO-SYMBOLIC APPROACH FOR BONGARD-OPENWORLD.

Bongard-OpenWorld is indeed built upon logical operations over visual concepts, which also serve as the foundation for BPs. Inspired by how our humans approach Bongard Problems, we design a neuro-symbolic approach that combines logical reasoning with VLMs & LLMs. Specifically, after obtaining the captions from a VLM, we leverage GPT-4 to generate meaningful concepts, which currently stands as the most powerful method for semantic understanding. Subsequently, we perform logical operations on the collection of concepts. Then, we update each negative concept with the intersection of positive images and each positive concept with the union of negatives. The intersection of positives must appear in at least four (positives total is six) pictures, although it may be absent in some pictures while still being part of the ground truth concepts. The union of negatives must appear in at least two (negatives total is six) pictures and may partially overlap with positives. We iterate this process until no further updates are required for each image, resulting in the most comprehensive information. Finally, we evaluate the presence of each intersection concept in the query image. If all the concepts exist, indicating that the intersection belongs to the query, it is considered positive; otherwise, it is deemed negative. Please refer to Algorithm 2 in the Appendix C for detailed implementation.

## 4 EXPERIMENTS

### 4.1 SETUP

We benchmark the approaches described in Section 3 to evaluate their performances on Bongard-OpenWorld. As we mentioned before, all the few-shot learner (excluding ChatGPT and GPT-4) adopts a ConvNext-base (Liu et al., 2022) image encoder. We consider four pretraining strategies: 1) no pretraining at all (scratch); 2) pretraining with ImageNet-1K dataset (IN-1K) (Deng et al., 2009); 3) pretraining with the full ImageNet dataset with more object categories (IN-22K); 4) pretraining with LAION-2B (Cherti et al., 2022; Schuhmann et al., 2022), a massive image-text collection (OpenCLIP). During training, the image encoder will be fine-tuned along with the few-shot learner regardless of being pretrained or not, but we use a smaller learning rate for the pretrained image encoders. For the auxiliary task, we connect the image encoder to the caption decoder of a pretrained `BLIP-2-opt-6.7B` model. Note both the QFormer and the query tokens are pretrained and a trading-off weight of $\alpha = 1.0$ is used. For LLM-based methods, we use the same BLIP-2 and InstructBLIP model to produce the captions for each image. Thanks to the large context length

Table 3: **Quantitative results on Bongard-OpenWorld.** All the non-LLM models use a ConvNeXt-base (Liu et al., 2022) image encoder, and we experiment with different pretraining strategies: no pretraining at all (scratch), pretraining with ImageNet-1K labels (IN-1K) (Deng et al., 2009), pretraining with full ImageNet-22K labels (IN-22k) and pretraining with LAION-2B (Schuhmann et al., 2022; Cherti et al., 2022) dataset (OpenCLIP). The framework corresponds to the four families of approaches depicted in Figure 3, w/ $\mathcal{T}$ and w/o $\mathcal{T}$ indicate whether the training set of Bongard-OpenWorld is utilized or not. While the LLM-based models use either BLIP-x or ChatCaptioner captions as the image representations. For the auxiliary captioning task, the few-shot learners are connected to the caption decoder of a pretrained `BLIP-2-opt-6.7B` model, and zero-shot models output captions by their reasoning. *denotes commonsense. **involves utilizing the ground truth concepts from Bongard-OpenWorld training set and the captions from BLIP-2 as inputs to fine-tuning ChatGPT over 5 epochs. The fine-tuned model is evaluated on the test set. It is worth noting that InstructBLIP was not fine-tuned due to a significant drop in its performance on ChatGPT. We splice raw images together in a manner similar to the examples in Appendix H, using only one query image each time, which is then inputted for GPT-4V reasoning.

| method | framework | image representation | aux. task? | splits | | | | avg. |
| | | | | short concept | long concept | CS* concept | anything else* | |
|---|---|---|---|---|---|---|---|---|
| Meta-Baseline | (a) w/ $\mathcal{T}$ | IN-22K | ✗ | 59.6 | 52.7 | 55.5 | 56.9 | 56.5 |
| | (a) w/ $\mathcal{T}$ | OpenCLIP | ✗ | 57.8 | 52.5 | 53.6 | 55.9 | 55.3 |
| | (a) w/ $\mathcal{T}$ | OpenCLIP | ✓ | 54.6 | 58.2 | 55.5 | 56.6 | 56.3 |
| MetaOptNet | (a) w/ $\mathcal{T}$ | scratch | ✗ | 52.3 | 51.6 | 54.5 | 51.0 | 52.0 |
| | (a) w/ $\mathcal{T}$ | IN-1K | ✗ | 60.6 | 47.3 | 54.5 | 54.5 | 54.5 |
| | (a) w/ $\mathcal{T}$ | IN-22K | ✗ | 61.5 | 51.5 | 53.6 | 57.9 | 56.8 |
| | (a) w/ $\mathcal{T}$ | OpenCLIP | ✗ | 63.3 | 51.6 | 50.9 | 60.7 | 58.0 |
| | (a) w/ $\mathcal{T}$ | OpenCLIP | ✓ | 62.8 | 51.1 | 51.8 | 59.7 | 57.5 |
| ProtoNet | (a) w/ $\mathcal{T}$ | scratch | ✗ | 57.8 | 50.5 | 48.2 | 56.9 | 54.5 |
| | (a) w/ $\mathcal{T}$ | IN-1K | ✗ | 56.9 | 54.9 | 51.8 | 57.6 | 56.0 |
| | (a) w/ $\mathcal{T}$ | IN-22K | ✗ | 62.4 | 51.6 | 54.5 | 58.6 | 57.5 |
| | (a) w/ $\mathcal{T}$ | OpenCLIP | ✗ | 61.9 | 53.8 | 59.1 | 57.9 | 58.3 |
| | (a) w/ $\mathcal{T}$ | OpenCLIP | ✓ | 59.2 | 57.7 | 51.8 | 61.0 | 58.5 |
| SNAIL | (a) w/ $\mathcal{T}$ | scratch | ✗ | 52.8 | 46.2 | 50.9 | 49.3 | 49.8 |
| | (a) w/ $\mathcal{T}$ | IN-1K | ✗ | 61.5 | 54.9 | 48.2 | 62.4 | 58.5 |
| | (a) w/ $\mathcal{T}$ | IN-22K | ✗ | 62.8 | 57.7 | 54.5 | 62.8 | 60.5 |
| | (a) w/ $\mathcal{T}$ | OpenCLIP | ✗ | 64.2 | 57.7 | 57.3 | 62.8 | 61.3 |
| | (a) w/ $\mathcal{T}$ | OpenCLIP | ✓ | 66.1 | **61.5** | **63.6** | 64.1 | **64.0** |
| OpenFlamingo | (b) w/o $\mathcal{T}$ | OpenCLIP | ✓ | 50.0 | 48.4 | 50.9 | 48.6 | 49.3 |
| Otter | (b) w/o $\mathcal{T}$ | OpenCLIP | ✓ | 49.3 | 49.3 | 48.9 | 49.4 | 49.3 |
| ChatGPT | (b) w/o $\mathcal{T}$ | BLIP-2 | ✓ | 60.6 | 56.6 | 55.5 | 60.0 | 58.8 |
| | (b) w/o $\mathcal{T}$ | InstructBLIP | ✓ | 52.1 | 50.6 | 48.1 | 52.7 | 51.4 |
| | (c) w/o $\mathcal{T}$ | ChatCaptioner | ✓ | 52.3 | 45.6 | 57.3 | 46.2 | 49.3 |
| ChatGPT (Fine-tuned)** | (b) w/ $\mathcal{T}$ | BLIP-2 | ✓ | 67.0 | 58.8 | 55.5 | **66.2** | 63.3 |
| GPT-4 | (b) w/o $\mathcal{T}$ | BLIP-2 | ✓ | 64.5 | 58.0 | 57.3 | 63.2 | 61.6 |
| | (b) w/o $\mathcal{T}$ | InstructBLIP | ✓ | **67.3** | 59.7 | 59.3 | 65.6 | 63.8 |
| GPT-4V | (b) w/o $\mathcal{T}$ | Raw Images | ✓ | 54.6 | 53.3 | 50.9 | 55.2 | 54.0 |
| Neuro-Symbolic | (d) w/o $\mathcal{T}$ | InstructBLIP | ✓ | 58.3 | 52.2 | 56.4 | 55.2 | 55.5 |
| Human | N/A | N/A | N/A | 91.7 | 90.1 | 89.1 | 91.7 | 91.0 |

allowed by the GPT-x family, no downsampling of captions is needed. Finally, we select the best model on the validation set and report its metrics on the test set proposed in Section 2.3. Full details on the hyperparameters can be found in the Appendix D.

## 4.2 ANALYSIS AND INSIGHTS

We provide the full quantitative results of the considered models (detailed in Section 3) on Bongard-OpenWorld in Table 3. The major findings are summarized below:

**Challenges of free-form visual concepts.** In Table 3, we demonstrate both overall averaged accuracy on the test set and also four types of splits as introduced in Section 2.3. It can be observed that once a model does better than coin-flipping (50% chance of success), it generally struggles more on Bongard-OpenWorld problems with longer free-form concepts versus shorter ones. Also, problems with abstract commonsense visual concepts (as described in Section 2.1) impose a greater challenge than those without it. This aligns with our hypothesis that few-shot reasoning with free-form visual concepts is indeed more difficult, compared to counterparts with a fixed set of image categories or HOIs, which are relatively short and likely do not include any commonsense aspects.

**Representations and open vocabulary.** As we mentioned before, reasoning with Bongard-OpenWorld problems requires the models to embrace an *open vocabulary* setting, where the set of all possible visual concepts are not provided a *priori*. Our experiments with different image representations verify that, by making the pretraining strategy closer to open vocabulary, *i.e.* expanding the vocabulary of visual concepts the image encoder is exposed to (in our case, from none to IN-1K, IN-22k, and LAION-2B label set), most of the evaluated few-shot learners can attain consistent improvement. Note that even with the fully open vocabulary pretraining (OpenCLIP), the gap between the best model (SNAIL) and humans is still quite substantial, implying that better pretraining is not a complete solution but more of a foundation for solving the reasoning task in Bongard-OpenWorld.

**The role of captioning.** In Table 2, we have shown that the free-form visual concepts in Bongard-OpenWorld are image captions. Therefore, we are interested in exploring to which extent the captioning task itself can help with Bongard-OpenWorld. First of all, we can observe that adding the auxiliary captioning task can slightly boost ProtoNet (+0.2%) while escalating Meta-Baseline (+2.5%) and SNAIL (+2.7%) more significantly. We also find fine-tuning with the auxiliary task could be more challenging as a large model (BLIP-2 decoder) is plumbed to the learner, making some few-shot learners unable to benefit from the auxiliary task.

On the other hand, even with off-the-shelf captions from BLIP-2, LLM-based methods still fail to close the human-machine gap. We hypothesize the reason to be two-fold: 1) *distractors*, without awareness of the Bongard-OpenWorld task and other images in the current problem, the captions BLIP-2 produces could be distracted by irrelevant content and miss description needed by the visual concepts; 2) *hard negatives*, as illustrated in Figure 1, the conceptual similarity between positives and negatives could perplex the concept induction of the LLM. Due to space limitation, we defer qualitative results of explicit concept induction of LLM-based method to the Appendix G.

**Different few-shot learners.** Many previous benchmarks (Nie et al., 2020; Jiang et al., 2022) have demonstrated that some few-shot learners are generally better than others in few-shot visual reasoning. Our winner here is SNAIL, a memory-based learner with a transformer architecture. It even surpasses the strong GPT-4 baseline on all measures. We hypothesize that the memory-based approach could suit Bongard-OpenWorld better as our dataset does not offer a huge amount of few-shot training episodes. Therefore, additional inductive biases like the sample retrieval in SNAIL could make the learner more data efficient.

**Limitations of current VLMs.** The results in Table 3 indicate that combining VLMs & LLMs in a zero-shot fashion, specifically using InstructBLIP to generate more complex captions, leads to a significant drop in the performance of ChatGPT. We hypothesize this drop is due to InstructBLIP introducing excessive noise that interferes with the reasoning of ChatGPT, while the robustness of GPT-4 remains unaffected. We attribute this to the limitations of current caption model in accurately representing open vocabulary free-form visual concepts, abstract visual attributes, and commonsense factual knowledge in Bongard-OpenWorld. Additionally, even powerful end-to-end pre-trained VLMs (e.g., OpenFlamingo, Otter) still face challenges with multi-image reasoning, which is a unique and particularly difficult aspect of Bongard-OpenWorld.

**Neuro-Symbolic also failed.** While performing logical operations directly on visual concepts may initially appear as an ideal solution, but the results are not satisfactory. GPT-4's ability to comprehend and extract concepts is unquestionable. However, we observed that the accuracy of inducing the true concept is disappointingly low, with a significant presence of irrelevant noise concepts, due to the inability of the concept extractor, *i.e.* the VLM. This leads us to point that, as mentioned earlier, developing a powerful model capable of multi-image reasoning and accurately inducing open vocabulary free-form visual concepts still requires tremendous effort.

## 5 CONCLUSION

We have introduced Bongard-OpenWorld, a benchmark for evaluating real-world few-shot reasoning for machine vision. It combines the best of few-shot learning and complicated real-world visual concepts. The diverse nature of these concepts including abstract visual attributes and commonsense factual knowledge impose great challenges to AI. State-of-the-art few-shot learners and even sophisticated systems reconcile LLMs & VLMs, largely fall behind human contestants. We invite people from different AI communities to join our challenge for this grand challenge.

## 6 Acknowledgments

We extend our heartfelt gratitude to the anonymous reviewers whose dedication and insightful feedback have significantly enhanced the caliber of this paper. Their constructive critiques and valuable suggestions were instrumental in refining our work. Additionally, we are deeply appreciative of the Program Chairs and Area Chairs for their meticulous handling of our submission and for their comprehensive and invaluable feedback. Their guidance has been pivotal in elevating the quality of our research.

This work is supported by National Key R&D Program of China (2022ZD0114900) and in part by National Natural Science Foundation of China (Grant No.61976214).

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

## A    LIMITATION STATEMENT

We carefully curate Bongard-OpenWorld to encompass open vocabulary and free-form visual concepts, which present exceptional challenges for current VLMs. We have investigated a range of approaches, including methods based on CLIP pre-training and ConvNeXt-base image encoder, combining pre-trained caption models with the GPT-x family, as well as incorporating interactive question-answering systems in single-round or multi-round fashion, even the neuro-symbolic approach performs logical operations on visual concepts. However, all of these attempts have yielded unsatisfactory results, and we have not discovered a promising solution. Nevertheless, we remain dedicated to further exploration and experimentation. For now, we propose that models capable of simultaneous multi-image reasoning hold the potential to tackle Bongard-OpenWorld challenge.

## B    BROADER DISCUSSION

We would like to emphasize that, adding open-world concepts is the basic requirement of a few-shot visual reasoning benchmark can be still relevant in the VLM era today. However, being able to handle open-world concepts like Flamingo/ChatGPT/Otter does not mean the model can solve Bongard-OpenWorld. Because there is one thing that could still be missing in these models: **reasoning-aware perception and holistic perception-reasoning**.

- **Separately done perception and reasoning cannot beat Bongard-OpenWorld, no matter how powerful the models are**.

Our benchmark requires extracting common visual concepts globally, rather than extracting atomic concepts from each image separately. The idea is, images in Bongard-OpenWorld have rich details, and therefore it is almost impossible to convert everything the images depicted into text and let LLM or other logic reasoner analyze what is being shared by positives and not depicted by negatives. Instead, VLM-based captioners usually fail to capture the image contents relevant to the current Bongard Problems (see Figure 5 in Appendix G, VLM is distracted by the cardinal and therefore did not describe the concept that is useful for reasoning, which is "tree or branch covered in snow"), therefore leading the follow-up reasoning with LLM fail.

- **Holistic perception-reasoning is hard, and it is true for largely pretrained VLMs (as for autumn 2023)**.

An ideal model for Bongard-OpenWorld will need to compare images from both positives and negatives, then figure out what is the right concept for classifying query image, i.e. reasoning-aware perception or holistic perception and reasoning. We, therefore, try our best to find an open-sourced VLM that has claimed such capability. Please note that the basic requirement will be handling multiple images at the same time, which already rules out most of the publically available VLMs. We ends up trying OpenFlamingo, Otter and GPT-4V. Clearly, even the latest and most powerful GPT-4V failed to close the human-machine gap as well.

The above reasons indeed highlight what might be going wrong with ChatGPT, which relies on a seperate VLM captioner, and Flamingo/Otter/GPT-4V, which could fall short on perform the reasoning required by Bongard-OpenWorld. On the contrary, all the few-shot learning baselines (from ProtoNet to SNAIL) have been trained on the training set of Bongard-OpenWorld, therefore they have learned (not perfect though) to perform the reasoning needed here, while also equipped with open-world or open-vocabulary perception thanks to the OpenCLIP backbone. Therefore, we can see the gap between the best FSL method and these zero-shot large models.

## C    MORE DETAILS ON BUILDING BONGARD-OPENWORLD

**Grid-sampling of visual concepts from CC-3M.**  We streamline all the captions from CC-3M into concept tuples and perform **grid sampling**. Its key operations to extract visual concepts from the streamlined captions and pseudo code as shown in Algorithm 1: 1) We run part-of-speech tagging (POST) and sliding window with size 2/3/4/5 over all the concept tuples to count the frequency of concept with length 2/3/4/5; 2) Instead of picking the top concept tuples of the whole CC-3M dataset,

---

**Algorithm 1** Grid sampling visual concepts $\mathcal{C}$ from CC3M

---

**Input:** CC-3M caption grids $Grids$, candidate lexicals $Lexs$, top factor $k$.
**Output:** Candidates for concept tuples $Q$.
 1: $Q = \emptyset$
 2: **for** $i = 1$ to $len(Grids)$ **do**
 3:    $grid = Grids[i]$;
 4:    $q = \emptyset$;
 5:    **for** $j = 1$ to $len(grid)$ **do**
 6:       $caption = grid[j]$;
 7:       $tuple = POST(caption)$;
 8:       $tuple = filter(tuple, key = Lexs)$;
 9:       $tuple = sliding\_window(tuple)$;
10:       $q = q \cup tuple$;
11:    **end for**
12:    $q = topK(q, K = k)$;
13:    $Q = Q \cup q$;
14: **end for**
15: **return** $Q$;

---

each time we only compute the frequency within a grid with only 300 (grid size) CC-3M concepts, pick the top k (k=3) as candidates for concepts $\mathcal{C}$, then move on to the next grid. We find this balances the need for sampling top concepts and sample diversity, as the variance introduced by a small grid facilitates the sampling of more long-tailed concepts in a large dataset like CC-3M.

**Crowd-sourcing for challenging visual concepts.** During our early inspection, we observed that visual concepts out of CC-3M are mostly about object categories and their relatively simple attributes & relations. However, humans can understand more abstract concepts that require *commonsense factual knowledge*. Therefore, we further augment the visual concepts with these challenging concepts through crowd-sourcing. Specifically, the annotators are instructed to write visual concepts by following a predefined set of categories illustrated in Table 9, more examples of non-commonsense and commonsense visual concepts are shown, including the ratio of each concept category in Bongard-OpenWorld. They are also asked to combine these challenging concepts with those mined from CC-3M. Our experiments confirm that Bongard-OpenWorld problems with these commonsense concepts become more difficult to solve.

**Caption metrics.** We also introduce a set of metrics that measure the correctness of the visual concepts explicitly induced by the model. As we mentioned before, the model will produce a sentence, *i.e.* image caption, which will then be compared against the ground truth. We use standard metrics for captioning tasks, including $BLEU_{1/2/3/4}$ (Papineni et al., 2002), METEOR (Banerjee & Lavie, 2005), $ROUGE_L$ (Lin, 2004) and CIDEr (Vedantam et al., 2015).

**Explicit caption induction.** These caption metrics provide objective measurements for assessing the quality and effectiveness of machine translation, text summarization, and image captioning systems. In our experiments, we evaluate two publicly available LLMs: ChatGPT and GPT-4. Since the public version of both models does not support multi-modal input, *e.g.* images, we implement the representation function repr($\cdot$) above as *image captioning*. Specifically, we ask BLIP-2 and InstructBLIP to produce the most-probable caption for all images in $\mathcal{S} = \mathcal{P} \bigcup \mathcal{N}$ and the query $I_q$. We also consider a variant that uses the URL of an image, which could leak some information about its content, as the representation. Since we are particularly interested in the visual concepts induced by the models, we further prompt the LLM to produce an explanation of the binary prediction it makes. Specifically, the model is guided to summarize the visual concepts that are exclusively depicted by the positives. The model's summary $\hat{\mathcal{C}}$ will then be compared against the ground truth $\mathcal{C}$ and evaluated using the image captioning metrics we introduced above.

**Logical operations on visual concepts.** We have presented a comprehensive description of logical operations in Section 3.4, the query concept will be updated by final intersection of positives.

---

**Algorithm 2** Logical operations on visual concepts

---

**Input:** Positive concepts $Cpt_{Positive}$, Negative concepts $Cpt_{Negative}$, Query concepts $Cpt_{Query}$.
**Output:** Intersection of positive concepts $Cpt_{Intersection}$.

1: **while** $True$ **do**
2:     $Cpt_{Intersection} = \bigcap_{i=1}^{6} Cpt_{Positive}^{i}$;
3:     $Cpt_{Intersection} = filter(Cpt_{Intersection}, value >= 4)$;
4:     $Cpt_{Union} = \bigcup_{i=1}^{6} Cpt_{Negative}^{i}$;
5:     $Cpt_{Union} = filter(Cpt_{Union}, value >= 2)$;
6:     $Cpt_{Positive}^{diff} = Cpt_{Union} - Cpt_{Positive}$;
7:     $Cpt_{Positive} = update(Cpt_{Positive}, Cpt_{Positive}^{diff})$;
8:     $Cpt_{Negative}^{diff} = Cpt_{Intersection} - Cpt_{Negative}$;
9:     $Cpt_{Negative} = update(Cpt_{Negative}, Cpt_{Negative}^{diff})$;
10:     **if** $Cpt_{Positive}^{diff} = \emptyset$ & $Cpt_{Negative}^{diff} = \emptyset$ **then**
11:       $break$;
12:     **end if**
13: **end while**
14: $Cpt_{Query}^{diff} = Cpt_{Intersection} - Cpt_{Query}$;
15: $Cpt_{Query} = update(Cpt_{Query}, Cpt_{Query}^{diff})$;
16: **return** $Cpt_{Intersection}$;

---

**Exact instructions for annotators.** When annotators are instructed to write visual concepts by following a predefined set of categories illustrated in Table 9 and asked to combine these challenging concepts with those mined from CC-3M, here are the guides to our annotators:

• General concept annotation guide:

1. Positive images all share one concrete concept, denoted as $\mathcal{C}$. This concept should be engaging and not overly simplistic (such as basic objects, 'dog' or 'cat'), but rather somewhat abstract (for instance, 'running' or 'reflection').

2. Each negative image must depict a concept that must have some added distractors with $\mathcal{C}$.

3. Each negative image must depict a concept that partially overlaps with $\mathcal{C}$.

• Commonsense annotation guide:

Follow the definition of commonsense, manually revise the automatically collected candidate concepts (from CC-3M) to make them more "commonsense"-alike.

One definition of commonsense from Wikipedia: "Commonsense knowledge includes the basic facts about events (including actions) and their effects, facts about knowledge and how it is obtained, facts about beliefs and desires. It also includes the basic facts about material objects and their properties." The following are some examples:

1. HOI: discerning subtle interaction between humans and objects, e.g. playing basketball vs. playing volleyball.

2. Taste / Nutrition / Food: identifying ingredients in food, e.g. high-calorie vs. high-sugar.

3. Color / Material / Shape: differentiating similar materials with various appearances, e.g. rough wood vs. painted wood.

4. Functionality / Status / Affordance: recognizing variants that have the same function, e.g. cuttable.

5. And / Or / Not: making logical judgments about something, e.g. no something.

6. Factual Knowledge: possessing factual knowledge, e.g. buildings in the capital of the United States.

7. Meta Class: induction of metaclass, e.g. rescue vehicle (including ambulance, police car, fire engine, etc), Felidae (including cat, lion, puma, etc).

8. Relationship: perceiving spatial relationships, e.g. "near", which may solely be "near on pixel distance" but located in the foreground and background respectively results in "far on physical distance".

9. Unusual observations: e.g. light refraction, gaze direction.

As you can see, the annotators are guided to revise & reorganize the original CC-3M concepts into commonsense concepts, with explicit guidance on what can be viewed as commonsense concepts. This aligns with the descriptions in the main paper.

## D  IMPLEMENTATION DETAILS AND HYPER-PARAMETERS

**Few-shot learning formulation.** Here we offer a formal definition of a few-shot learning task in Bongard-OpenWorld. Recall the formulation in Section 2, each of our Bongard problem instance $\langle \mathcal{C}, \mathcal{P}, \mathcal{N}, I_q \rangle$ comprises one few-shot learning *episode* with $N = 2$ classes and $2M$ samples. Therefore the learner has to learn from a set $\mathcal{S} = \mathcal{P} \bigcup \mathcal{N} = \{(I_1^P, 1), \ldots, (I_M^P, 1), (I_1^N, 0), \cdots, (I_M^N, 0)\}$ and is evaluated on a query image $(I_q, y_q)$. Each example $(I, y)$ includes an image $I \in \mathbb{R}^{H \times W \times 3}$ and a class label $y \in \{0, 1\}$, indicating if the visual concepts $\mathcal{C}$ (exclusively depicted by $\mathcal{P}$) presents in $I$. We make $M = 6$ in Bongard-OpenWorld and therefore it becomes a "2-way, 6-shot" few-shot learning episode.

**Distractors in positives and hard negatives.** To further increase the intra-diversity among the positives $\mathcal{P}$ and therefore perplex the induction of visual concepts, given $\mathcal{C}$, we prompt ChatGPT to expand it into 10 *sentences for the positives* by inserting "distracting" objects, attributes, etc. while ensuring their common ground is still $\mathcal{C}$. Moreover, prior work on Bongard Problems has shown that improper choices of the negative set $\mathcal{N}$, *e.g.* a set of randomly chosen images, could trivialize the challenge by allowing the model to merely recognize part of the visual concepts $c$ (*e.g.* only recognizing "animals" of a full concept "animals are running") and solve the problem. Therefore, the content of the negative images should overlap with $\mathcal{C}$ to ensure the need of inducing full visual concepts. To this end, we again prompt ChatGPT to edit $\mathcal{C}$ into 10 *sentences for the negatives* that only partially overlap with it. Both sets of sentences will then be used to collect images. For all models, we invoke the latest version from OpenAI and set "temperature" as 1.0.

We use the ChatCompletion API of `gpt-3.5-turbo` model and following prompt:

> Given a concept such as "concept: bird fly". Then, Please help me to complete the following 2 tasks:
> 1. Expand the "concept" into 10 positive sentences, their semantics are different but both share all words in the "concept". For example, positive sentences could be "positive: [a bird fly in the sky, a bird fly in the rainy day, the large bird fly above clouds, a bird fly over the river, a bird fly over mountains, some birds fly in a V formation, birds in a row flying in a colorful sky, bird fly icon, bird fly papercut, cartoon illustration of bird fly]".
> 2. Reduce the "concept" into 10 negative sentences, their semantics are different but both missing only one word in the "concept". For example, negative sentences could be "negative: [a bird eating seeds, a bird singing on a wire, a bird building a nest, a ladybug fly in the rainy day, a bird perched on a branch, a bird pecking at the ground, an airplane fly over mountains, a bird trapped in a cage, bird specimen, fly fish]".
>
> Please complete the following example:

**VLM + LLM for Bongard-OpenWorld.** We evaluate two publicly available LLMs: ChatGPT and GPT-4, both in single-round and multi-round fashion. We ask BLIP-2 and InstructBLIP to produce the most-probable caption for all images in $\mathcal{S} = \mathcal{P} \bigcup \mathcal{N}$ and the query $I_q$. For the multi-round fashion, we followed the ChatCaptioner (Zhu et al., 2023) prompt design.

For the single-round fashion, we use the ChatCompletion API of `gpt-3.5-turbo` and `gpt-4` model and following prompt:

> Given 6 "positive" sentences and 6 "negative" sentences, where "positive" sentences can be summarized as 1 "common" sentence and "negative" sentences cannot, the "common" sentence describes a set of concepts that are common to "positive" sentences. And then given 1 "query" sentence, please determine whether the "query" belongs to "positive" or "negative" and give the "common" sentence from "positive" sentences.
>
> Please complete the following query:

**End-to-end pre-trained VLMs for Bongard-OpenWorld.** In our VLMs experiments, we evaluate two publicly available VLMs: OpenFlamingo and Otter, we use the same prompt as following:

> Given 6 "positive" images and 6 "negative" images, where "positive" images share "common" visual concepts and "negative" images cannot, the "common" visual concepts exclusively depicted by the "positive" images. And then given 1 "query" image, please determine whether it belongs to "positive" or "negative".
> "positive" images:<|endofchunk|><image><image><image><image><image><image>
> "negative" images:<|endofchunk|><image><image><image><image><image><image>
> "query" image:<|endofchunk|><image>
> "query" image belongs to

**Neuro-symbolic approach for Bongard-OpenWorld.** We use GPT-4 to extract meaningful concepts from caption sentences, the prompt as following:

> Given a sentence that describes a set of visual concepts, these concepts may be Adjectives, Nouns, Verbs, or Adverbs. Please identify these concepts and organize them into a Python list.

**We employ GPT-4V for reasoning on legends similar to those in Appendix H, using a single query image for each instance, the prompt as following:**

> This image layout consists of three rows. The first row displays six positive images that all depict one shared visual concept, and the second row displays six negative images none of which depict this concept. The third row features a single query image.
> Please look very carefully for details of all images and perform contextual reasoning, determine whether query belongs to positive or negative, and summarize the shared visual concept in a sentence only includes unique core visual concepts.
> WARNING: the query belongs to the positive only when it depicts the shared concept; otherwise it belongs to the negative.
> WARNING: please look very carefully and reexamine your result before answering.
> positive or negative: the shared visual concept:

**Few-shot learning for Bongard-OpenWorld.** Main hyperparameters of each few-shot model in our experiments are shown in Table 8.

## E  ADDITIONAL EMPIRICAL RESULTS

### E.1  EXPLICIT VISUAL CONCEPTS INDUCTION ON BONGARD-OPENWORLD.

Due to space limitation, we defer quantitative results of explicit concepts induction of LLM-based method to here. We use the aforementioned caption metrics to measure the performance of LLM-based models on induction of visual concepts. As depicted in Table 4, GPT-4 exhibits not only 2.8% higher binary prediction accuracy compared to ChatGPT but also demonstrates a notable improvement in explicit visual concepts induction. We speculate that the performance of LLM may be detrimentally affected by the quality of captions generated by VLM.

Table 4: **Explicit visual concepts induction on Bongard-OpenWorld.** We prompt the LLM-based approaches to explicitly summarize the visual concepts of the positives and evaluate their output based on the captioning metrics introduced in Appendix C.

| method | representation | $BLEU_1$ | $BLEU_2$ | $BLEU_3$ | $BLEU_4$ | METEOR | $ROUGE_L$ | CIDEr |
|---|---|---|---|---|---|---|---|---|
| | BLIP-2 | 0.186 | 0.109 | 0.070 | 0.045 | 0.154 | 0.258 | 0.781 |
| ChatGPT | BLIP-2 w/ Fine-tuning | **0.441** | **0.292** | **0.209** | **0.153** | **0.222** | **0.417** | **1.714** |
| | InstructBLIP | 0.111 | 0.060 | 0.034 | 0.017 | 0.135 | 0.182 | 0.198 |
| GPT-4 | BLIP-2 | 0.310 | 0.199 | 0.140 | 0.100 | 0.207 | 0.358 | 1.351 |
| | InstructBLIP | 0.136 | 0.075 | 0.045 | 0.027 | 0.154 | 0.202 | 0.256 |
| GPT-4V | Raw Images | 0.190 | 0.073 | 0.029 | 0.023 | 0.111 | 0.188 | 0.527 |

## E.2 RESULTS COMPARED TO RELEVANT BENCHMARK (BONGARD-HOI).

Compared to Bongard-HOI, there are both shared observations and differences.

1. shared observations: canonical FSL (ex. SNAIL, MetaBaseine) do not work well on both datasets, regardless what representation backbone is being used.

2. differences:

- the most effective methods differ (Bongard-OpenWorld is SNAIL vs. Bongard-HOI is Meta-Baseline);

- Bongard-HOI is more sensitive to open-vocabulary pretrained representations (its concept vocabulary is limited); Bongard-OpenWorld is not as it is designed to be open vocabulary or open-world;

- Bongard-HOI does not have results of LLM-based methods.

The first and last observations are obvious. Here we would like to elaborate a bit on the second one. We offered some additional results of benchmarking CLIP (shown below), a relatively more free-form and open-world visual backbone on both Bongard-HOI (a prior work, focus on compositional generalization on closed-world concepts) and Bongard-Openworld. As you can see, with a much stronger backbone like OpenCLIP, ProtoNet can achieve 71.3% acc on Bongard-HOI (way higher than the previous Bongard-HOI SOTA 58.3% using ImageNet-1k pretrained backbone), but on Bongard-OpenWorld it can only reach 58.3%. In fact, Bongard-HOI discourages the use of such an open-world pretrained backbone like CLIP because it might undermine the challenges posed by the benchmark (compositional generalization will not be an issue anymore, as all concepts in training/generalization test have been saw by CLIP). Rather, Bongard-OpenWorld allows and indeed encourages all types of backbone and models by introducing free-form open-world concepts. And as the additional result suggest, we're successful – even the powerful CLIP backbone is still far from beating our benchmark.

Table 5: **Results compared to Bongard-HOI.** With a much stronger backbone like OpenCLIP, ProtoNet can achieve 71.3% acc on Bongard-HOI (way higher than the previous Bongard-HOI SOTA 58.3% using ImageNet-1k pretrained backbone), but on Bongard-OpenWorld it can only reach 58.3%.

| benchmark | method | image representation | aux. task? | avg. |
|---|---|---|---|---|
| Bongard-OpenWorld | ProtoNet | ImageNet-1k | ✗ | 56.0 |
| Bongard-OpenWorld | ProtoNet | OpenCLIP | ✗ | 58.3 |
| Bongard-HOI | ProtoNet | ImageNet-1k | ✗ | 58.3 |
| Bongard-HOI | ProtoNet | OpenCLIP | ✗ | 71.3 |

## E.3 IMBALANCE BETWEEN POSITIVE AND NEGATIVE QUERIES.

This section presents further results singling out the accuracy on positive and negative queries. Table 6 shows that there might be slight difference for a single model, but we did not observe a pattern or preference towards positive or negative queries globally. Therefore Bongard-OpenWorld does not appear to exhibit a strong imbalance issue.

Table 6: **Results of imbalance between positive and negative queries.** Each result is derived from the most effective model corresponding to each method.

| method | image representation | aux. task? | positive query | negative query | avg. |
|---|---|---|---|---|---|
| Meta-Baseline | OpenCLIP | ✓ | 59.5 | 53.0 | 56.3 |
| MetaOptNet | OpenCLIP | ✓ | 57.9 | 57.0 | 57.5 |
| ProtoNet | OpenCLIP | ✓ | 58.2 | 58.7 | 58.5 |
| SNAIL | OpenCLIP | ✓ | 61.2 | 66.8 | 64.0 |
| GPT-4 | InstructBLIP | ✓ | 67.4 | 60.2 | 63.8 |

### E.4 RESULTS OF NATURAL BASELINE.

In fact, natural baseline could be very similar to ProtoNet (which we already benchmarked), but ProtoNet further fine-tunes the visual features on the training set. This is how we implement it: we calculates the mean similarity between a query image embedding (produced by CLIP or DINO) and both positive and negative images embeddings (also produced by CLIP or DINO) in the support set, assigning the label of the higher mean as its prediction.

Table 7 reveals that natural baseline with both CLIP and DINO as backbone fail to produce comparable performances to the existing baselines. Our hypothesis aligns with the initial analysis with CLIP, when constructing Bongard-OpenWorld, we deliberately increase the similarity between positive queries and negative images in order to make the problem harder. Although the similarity is measured with CLIP embedding disantance, as the results above suggested, embeddings based on other approaches, ex. DINO and DINO-v2 could also be affected.

Table 7: **Results of natural baseline.** The model architecture for CLIP is ViT-H/14, which reached a performance of 78.0% on the ImageNet top-1 accuracy. The model architecture for DINO is ViT-B/8, which achieved a performance of 80.1% on the ImageNet top-1 accuracy. The model architecture for DINOv2 is ViT-g/14 (with registers), achieving a performance of 87.1% on the ImageNet top-1 accuracy.

| method | image representation | aux. task? | short concept | long concept | CS* concept | anything else | avg. |
|---|---|---|---|---|---|---|---|
| natural baseline | OpenCLIP | ✗ | 12.8 | 7.7 | 12.7 | 9.7 | 10.5 |
| | DINO | ✗ | 18.3 | 11.0 | 16.4 | 14.5 | 15.0 |
| | DINOv2 | ✗ | 17.9 | 11.0 | 14.5 | 14.8 | 14.8 |
| ProtoNet | OpenCLIP | ✓ | 59.2 | 57.7 | 51.8 | 61.0 | 58.5 |

## F RELATED WORK

**Free-form and open vocabulary computer vision.** Historically, due to the limitation on data annotation and model training, many of the computer vision benchmarks only adopt a small set of visual concepts, *e.g.* PASCAL VOC (Everingham et al., 2010), Microsoft COCO (Lin et al., 2014) and even the most popular split of ImageNet (Deng et al., 2009) only includes 1K object categories. The recent surge of pretraining and open-ended models like LLMs (Devlin et al., 2018; Brown et al., 2020) has called for the free-form open-vocabulary vision models and benchmarks (Kuo et al., 2022; Kirillov et al., 2023; Zareian et al., 2021; Li et al., 2022; Radford et al., 2021). While the goal is to accommodate any visual concepts, most of these works so far only serve the task of mid or high-level vision, including language-driven image recognition and object detection/segmentation. Our Bongard-OpenWorld benchmark fills in the blank of open vocabulary in visual reasoning and few-shot learning. To solve our tasks, these models have to go beyond recognition, step into reasoning, and induce open-world visual concepts. Our results with state-of-the-art open vocabulary recognition model like BLIP-2 (Li et al., 2023) and CLIP (Radford et al., 2021) also confirms that Bongard-OpenWorld cannot be trivially solved by these models.

**Few-shot and meta learning methods.** Few-shot learning studies learning from limited number of examples (Fe-Fei et al., 2003; Zhu & Zhu, 2021). In the past decade, meta-learning (or learning-to-learn) (Hochreiter et al., 2001) has become a leading family of approaches to tackle the few-shot learning challenge. The main idea is to discover the generic knowledge across tasks to adapt to a new task quickly. Several meta-learning methods have been developed and can be categorized into three types: 1) memory-based methods, such as MANN (Santoro et al., 2016) and SNAIL (Mishra et al., 2018); 2) metric-based methods, such as Matching Networks (Vinyals et al., 2016) and Pro-toNet (Snell et al., 2017), and 3) optimization-based methods, such as MetaOptNet (Lee et al., 2019)

Table 8: **Hyperparameters for the few-shot learning models.** The MetaOptNet models shows experimental results for *scratch*, *IN-1K*, *IN-22K*, *OpenCLIP* and *OpenCLIP+Caption*. The hyperparameters of *scratch*, *IN-1K* and *IN-22K* options for other few-shot models are consistent with MetaOptNet, therefore all of them are omitted, showcased are their own unique hyperparameters.

| Parameter | Value |
|---|---|
| *MetaOptNet* | |
|     Optimizer | AdamW |
|     Learning rate | $1e^{-4}$ |
|     Weight decay factor | $5e^{-4}$ |
|     Learning rate schedule | Warmup cosine |
|     Total training epochs | 20 |
|     Batch size | 10 |
|     Image encoder learning rate | $1e^{-5}$ |
| *MetaOptNet (Caption Only)* | |
|     Caption head | OPT captioner |
|     Caption LM | facebook/opt-125m |
|     Use qformer | True |
|     Pretrained qformer | True |
|     Freeze qformer | False |
|     Pretrained query | True |
|     Freeze query | False |
|     Use unpooled | True |
|     Caption loss coefficient | 0.1 |
| *Meta-Baseline* | |
|     Image encoder learning rate | $1e^{-4}$ |
| *Meta-Baseline (Caption Only*)* | |
|     Freeze qformer | True |
|     Freeze query | True |
|     Caption loss coefficient | 0.1 |
| *ProtoNet* | |
|     Image encoder learning rate | $3e^{-5}$ |
| *ProtoNet (Caption Only)* | |
|     Freeze qformer | True |
|     Freeze query | True |
|     Caption loss coefficient | 1.0 |
| *SNAIL* | |
|     Image encoder learning rate | $1e^{-4}$ |
| *SNAIL (Caption Only)* | |
|     Freeze qformer | False |
|     Freeze query | False |
|     Caption loss coefficient | 1.0 |

and ANIL (Raghu et al., 2020). These approaches have achieved excellent performances on existing few-shot learning benchmarks such as miniImageNet (Vinyals et al., 2016) and Omniglot (Lake et al., 2015). However, these benchmarks are primarily focusing on image recognition with a fixed and relatively small set of visual concepts (mostly object categories), *i.e.* they fail to account for the challenging free-form and open vocabulary visual concepts in a few-shot learning setting. We believe our Bongard-OpenWorld can help fill this gap and foster the emergence of better few-shot learning methods for open-world free-form visual concepts.

**Visual reasoning benchmarks.** Abstract visual reasoning is believed to be a core capability of human-level visual intelligence (Marr, 2010; Zhu & Zhu, 2021; Ma et al., 2022a) and therefore several benchmarks have been developed to measure the progress of current AI on it. Notably examples include compositional question answering (Johnson et al., 2017; Yi et al., 2019), physical reasoning (Bakhtin et al., 2019), mathworld problems (Saxton et al., 2019) and general AI (Xie et al., 2021; Chollet, 2019). Some more relevant benchmarks also consider a few-shot reasoning setting as ours, including RPM (Barrett et al., 2018; Zhang et al., 2019; Teney et al., 2020), Bongard Problems with synthetic shapes (Nie et al., 2020), physical problems (Weitnauer & Ritter, 2012) and real-world images (Jiang et al., 2022). While the reasoning problems within most of these benchmarks only comprise a limited number of simple visual concepts like object categories and shapes, our Bongard-OpenWorld introduces a new challenge of reasoning about free-form open vocabulary concepts in a few-shot fashion. Moreover, the open-world nature of our benchmark invites any AI models with no limitation on pretraining, etc, making it more friendly to many recent AI advances.

## G  QUALITATIVE RESULTS

**VLM+LLM (single-round).** Here we only consider the GPT-4 + BLIP-2 model. The results echo our hypothesis in Section 4.2, when BLIP-2 can accurately capture the visual concepts, GPT-4 can perform perfect reasoning (as shown in Figure 4). However, when BLIP-2 misses some details (as shown in Figure 5), it pays more attention to the "bird" in the foreground and ignores the "tree branches covered in snow" in the background, resulting in GPT-4 making correct concept induction but fails on binary prediction.

**VLM+LLM (multi-round).** Interactive update information can solve this problem instead of using fixed captions, as described in Section 3.3. GPT-4 combines the captions of the query and support set images to conduct multi-round question-answering. This allows it to utilize more detailed and accurate information and perform perfect reasoning, as shown in Figure 6.

**Neuro-symbolic approach.** We invoke the logic reasoner (detailed in Algorithm 2) on GPT-4, which leads to accurate concepts being updated from fixed captions if they are missing, as shown in Figure 7.

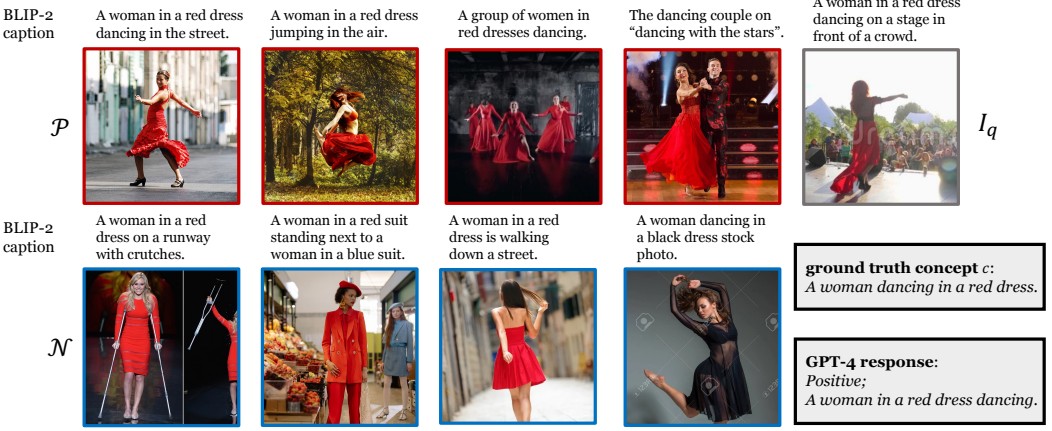

Figure 4: GPT-4 correctly produces both binary prediction and induced visual concepts.

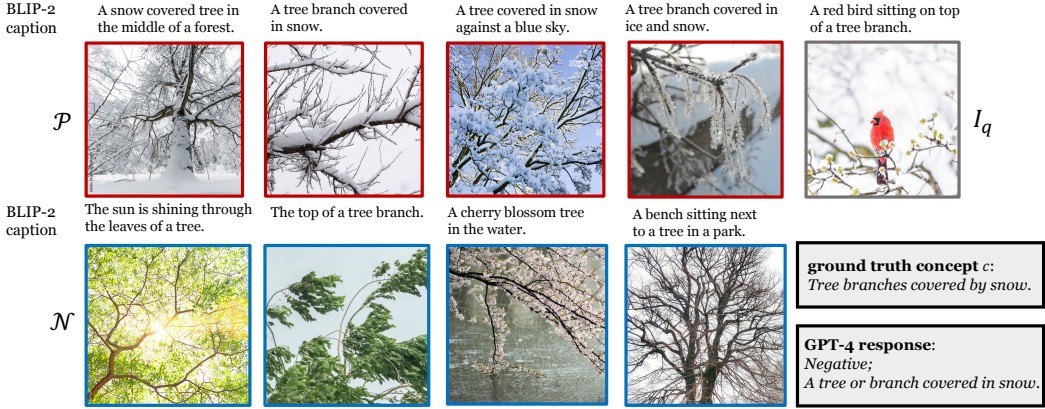

Figure 5: BLIP-2 only covers unhelpful content of $I_q$, GPT-4 makes correct concept induction but fails on binary prediction.

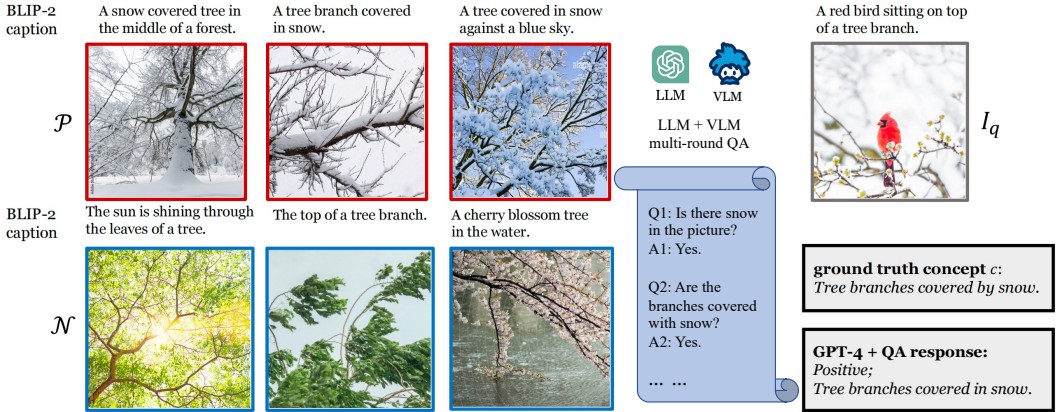

Figure 6: Multi-round question-answering corrects the missing details of images in fixed captions, allowing GPT-4 to execute correct reasoning.

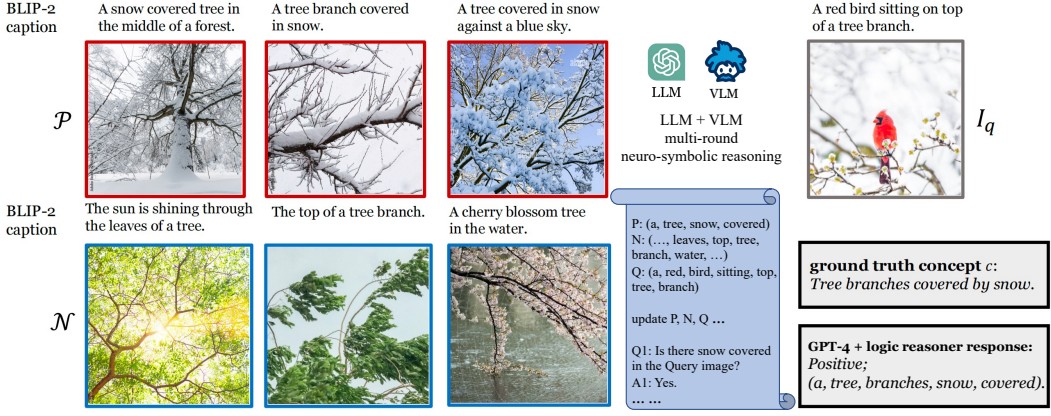

Figure 7: The logical reasoner corrects the missing details of images in fixed captions, enabling GPT-4 to perform perfect reasoning.

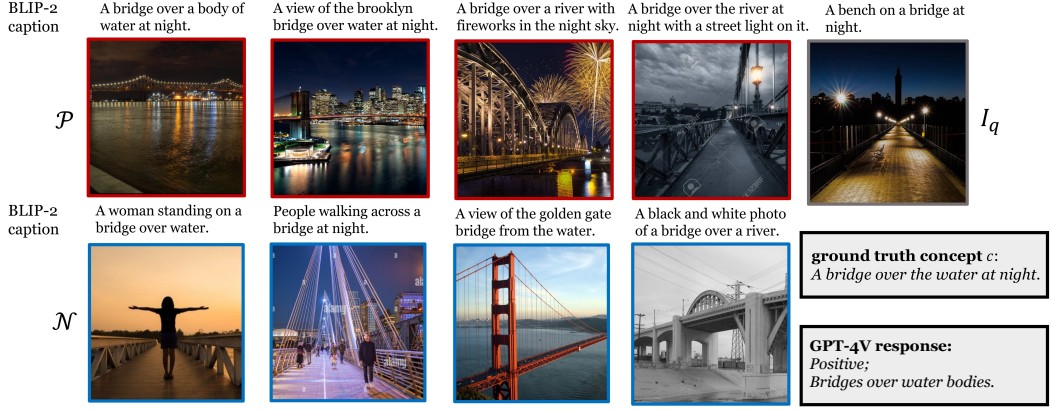

Figure 8: GPT-4V also experiences hallucinations during reasoning, which leads to difficulties in the conceptual induction of certain images (here is $I_q$), resulting in failures in binary prediction but remains correct in concept induction.

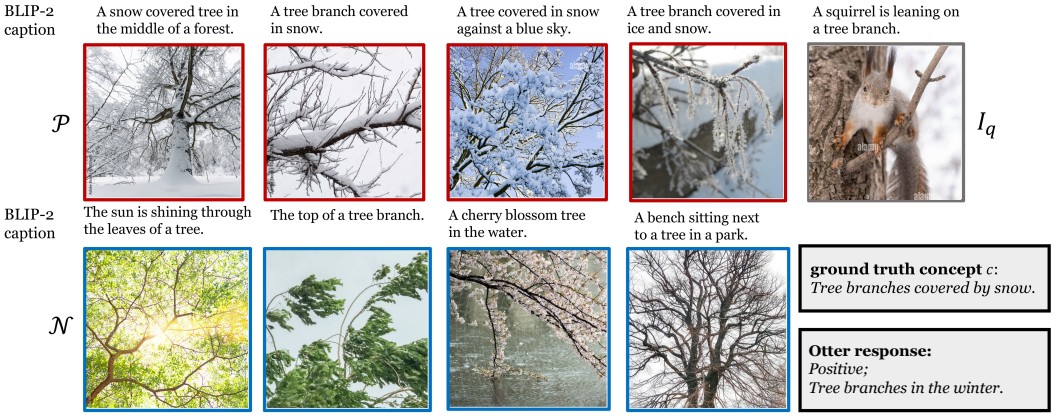

Figure 9: The hallucinations in Otter are more pronounced, leading to failures of both binary prediction and induced visual concepts.

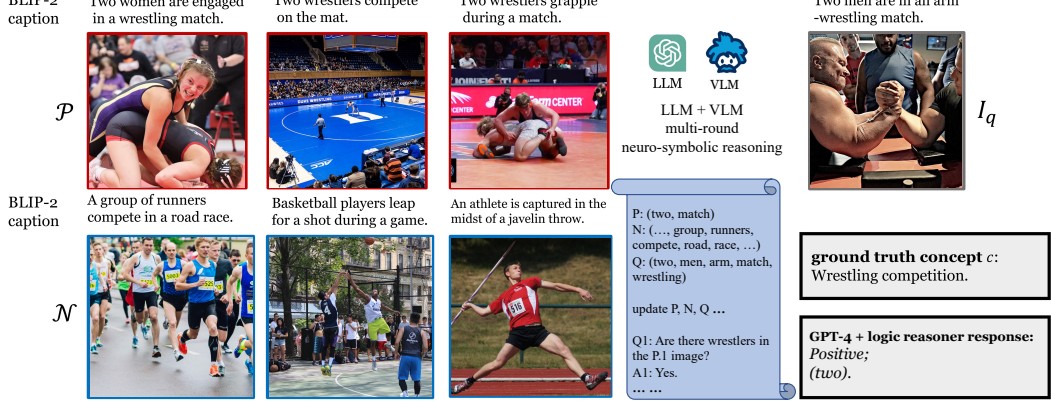

Figure 10: When concepts require a certain level of generalization (which is relatively easy for humans), the concepts extracted from their captions are too chaotic. Any update attempts of Neuro-Symbolic logical reasoner fail to yield favorable outcomes, consequently resulting in failures of both binary prediction and induced visual concepts.

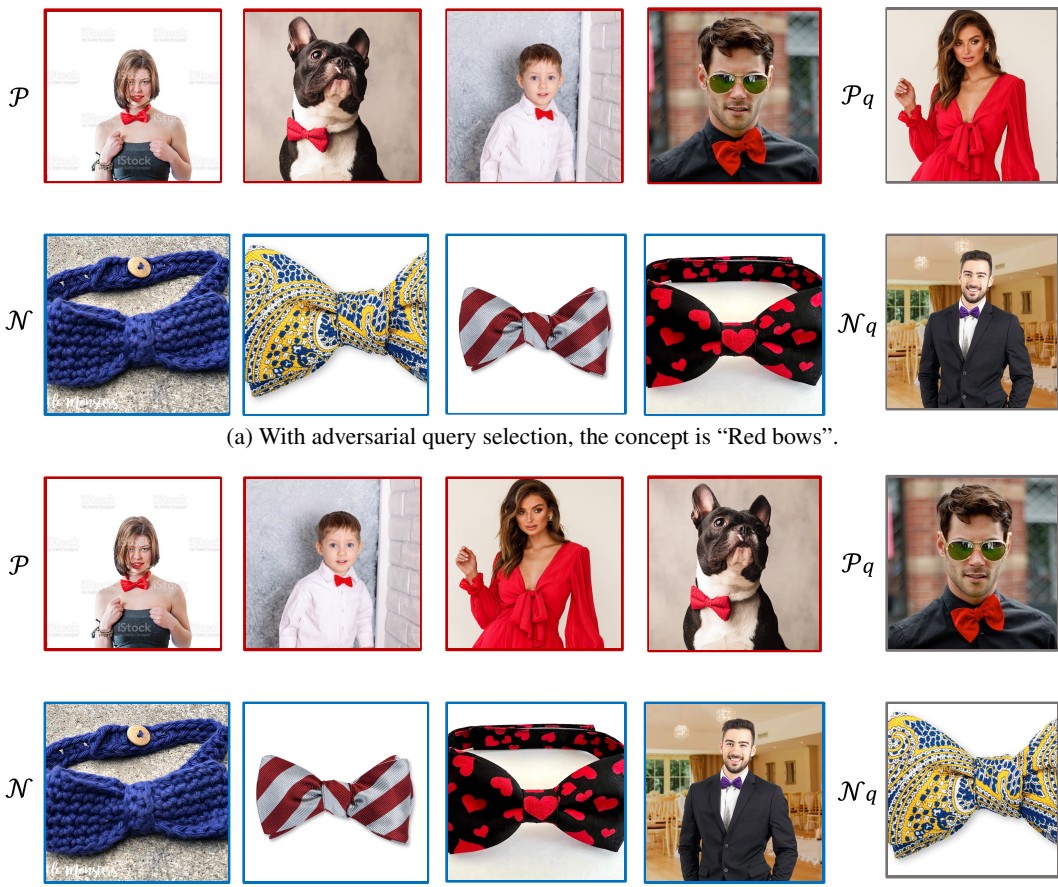

(a) With adversarial query selection, the concept is "Red bows".

(b) Without adversarial query selection, it is straightforward to find similar images in the support set.

Figure 11: Illustration to visually demonstrate the ablation analysis of design choices.

Table 9: **Examples of each category visual concepts in Bongard-OpenWorld**. We demonstrate the ratio of each concept category and more examples, including those mined from CC-3M (anything else) and challenging commonsense-related concepts that are crowd-sourced and augmented to the dataset.

| Concept Category | Ratio | ID | Example |
|---|---|---|---|
| Anything else | 73.4% | 0 | Strawberry leaves.
Kids playing in the river.
A perched mantis hanging on a plant.
Reeds swaying in the wind.
Cars on the city streets at night. |
| HOI | 2.7% | 1 | Persons riding bicycles.
Riders riding horses.
People holding the American flag.
A girl is holding a doll.
A farmer wearing a hat. |
| Taste / Nutrition / Food | 2.5% | 2 | Grilled steaks.
Spicy foods.
A bowl of chocolate pudding.
Some delicious food with shrimp meat.
A plate of vegetable salad. |
| Color / Material / Shape | 5.7% | 3 | Steel beams of the building.
The tungsten lamp is glowing.
Ceramic bowl.
Leather bags.
Bamboo baskets. |
| Functionality / Status / Affordance | 3.0% | 4 | Birds soaring in the air.
A colorful balloon floating in the air.
The faucet that is discharging water.
People cheering on the soccer field.
Small animals that can fly. |
| And / Or / Not | 1.5% | 5 | The room corner without people.
Wedding photos of the bride and groom.
A collection of seashells or conches.
A baboon and a car.
A stall selling vegetables and fruits. |
| Factual Knowledge | 2.8% | 6 | Capital of the US.
A distant view of the Egyptian pyramids.
The American flag.
Golden Gate Bridge.
Sydney Opera House. |
| Meta Class | 2.0% | 7 | Canine animals.
Marine animals swimming in the sea.
Military carriers.
Ball bats.
Rescue vehicles. |
| Relationship | 1.9% | 8 | Coral reefs on the bottom of sea.
A feather gently falls on the ground.
A town surrounded by mountains.
An airplane flying above clouds.
Books stacked on top of each other. |
| Unusual observations | 4.7% | 9 | Clear bottom of lake transmits the sunlight.
The moonlight reflected in the lake at night.
A Shimmering water surface.
Rusty metal hooks.
People are in a hurry. |

## H   MORE BONGARD-OPENWORLD EXAMPLES

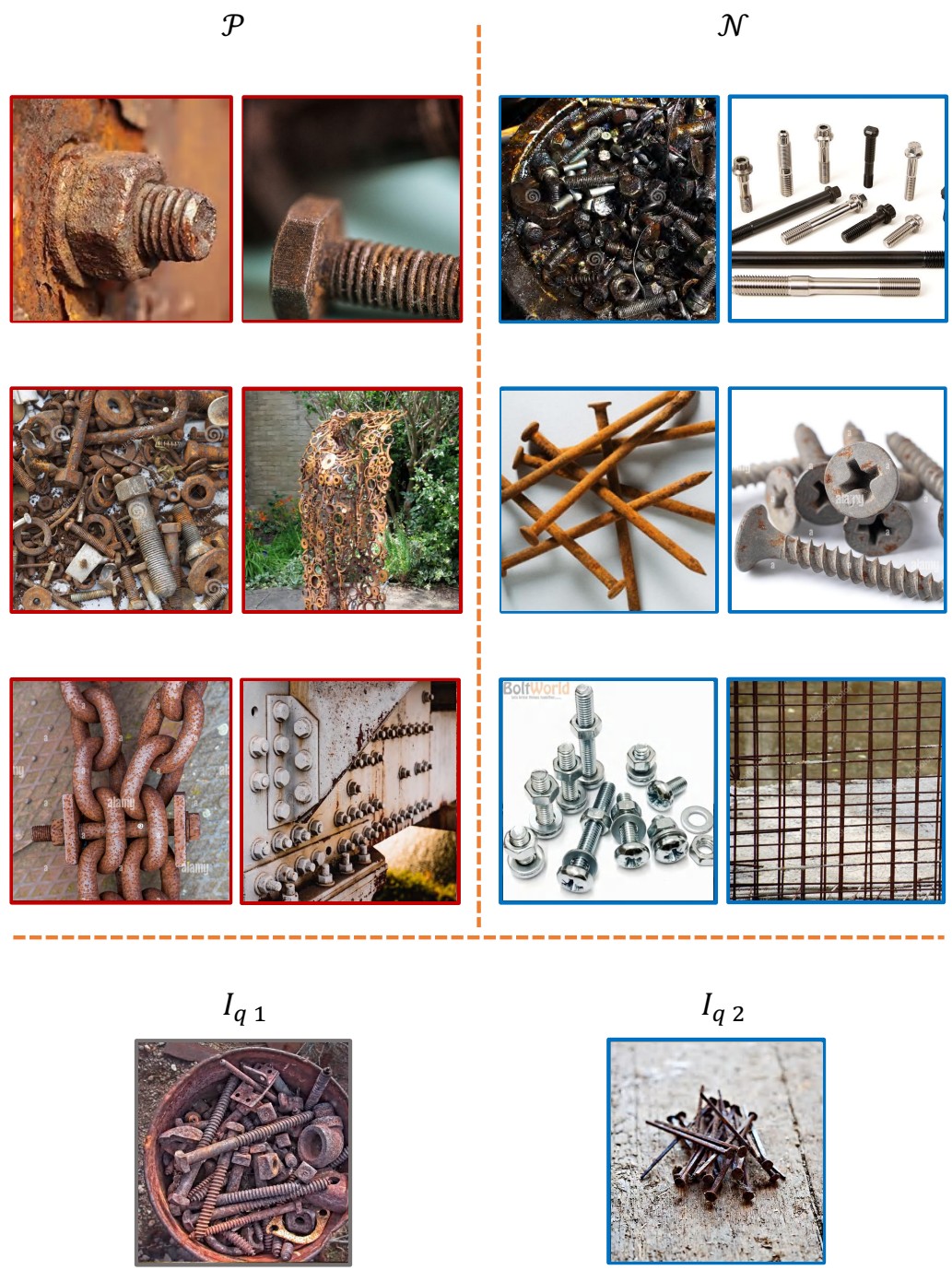

Figure 12: **Example of Bongard-OpenWorld.** The images in set $\mathcal{P}$ all depict *"rusty bolts"*, while the images in set $\mathcal{N}$ have *"rusty nails"*, *"rusty wires"* or others, so the concept $\mathcal{C}$ exclusively depicted by $\mathcal{P}$ is *"rusty bolts"*.

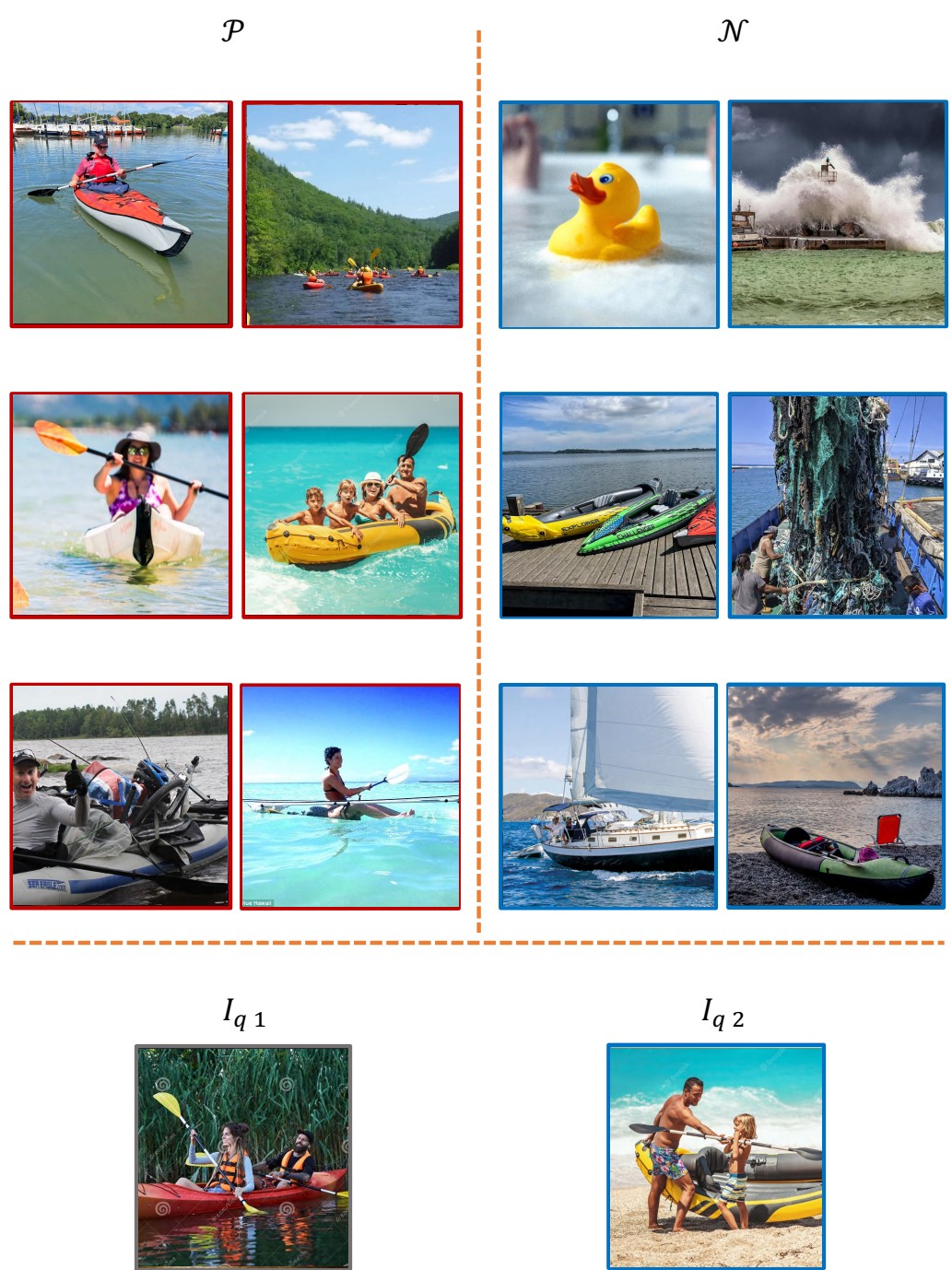

Figure 13: **Example of Bongard-OpenWorld.** The images in set $\mathcal{P}$ all depict *"rubber kayak in the water"*, while the images in set $\mathcal{N}$ have *"rubber kayak"*, *"boat in the water"* or others, so the concept $\mathcal{C}$ exclusively depicted by $\mathcal{P}$ is *"rubber kayak in the water"*.

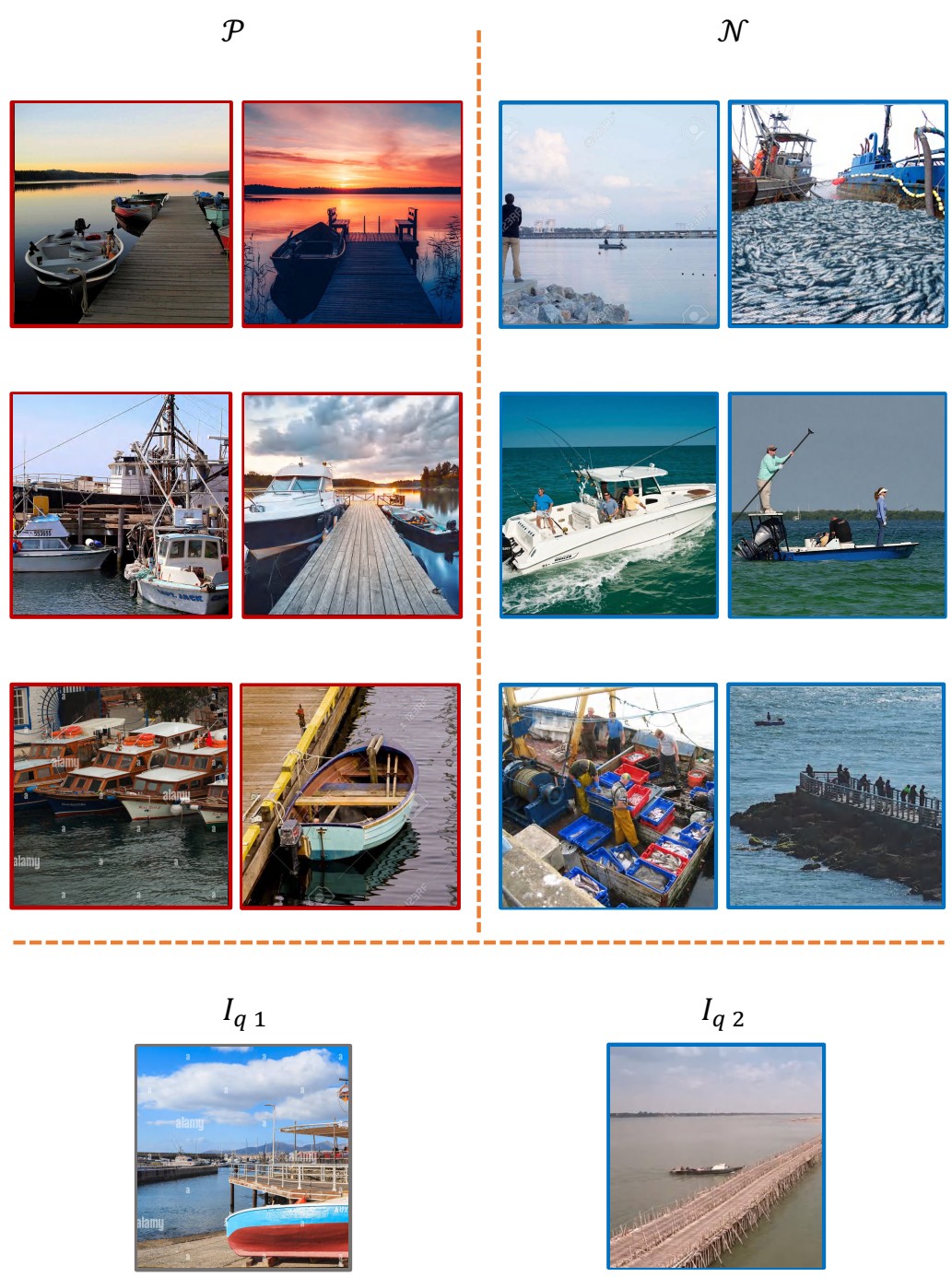

Figure 14: **Example of Bongard-OpenWorld.** The images in set $\mathcal{P}$ all depict *"boats docked at the pier"*, while the images in set $\mathcal{N}$ have *"boats in the ocean"*, *"a boat passing the pier"* or others, so the concept $\mathcal{C}$ exclusively depicted by $\mathcal{P}$ is *"boats docked at the pier"*.

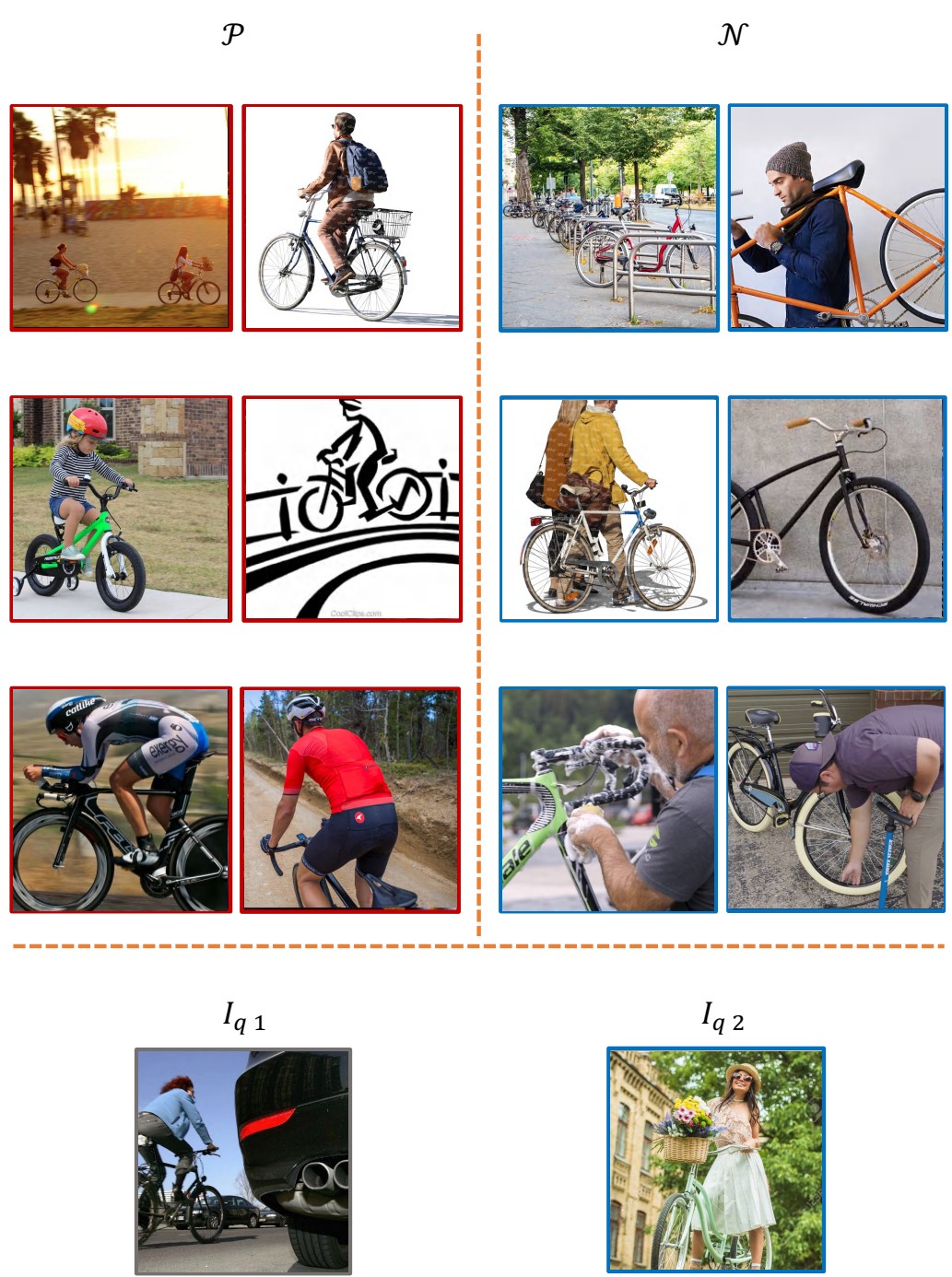

Figure 15: **Example of Bongard-OpenWorld.** The images in set $\mathcal{P}$ all depict *"person riding bicycle"*, while the images in set $\mathcal{N}$ have *"person pushing bicycle"*, *"person carrying bicycle"* or others, so the concept $\mathcal{C}$ exclusively depicted by $\mathcal{P}$ is *"person riding bicycle"*.

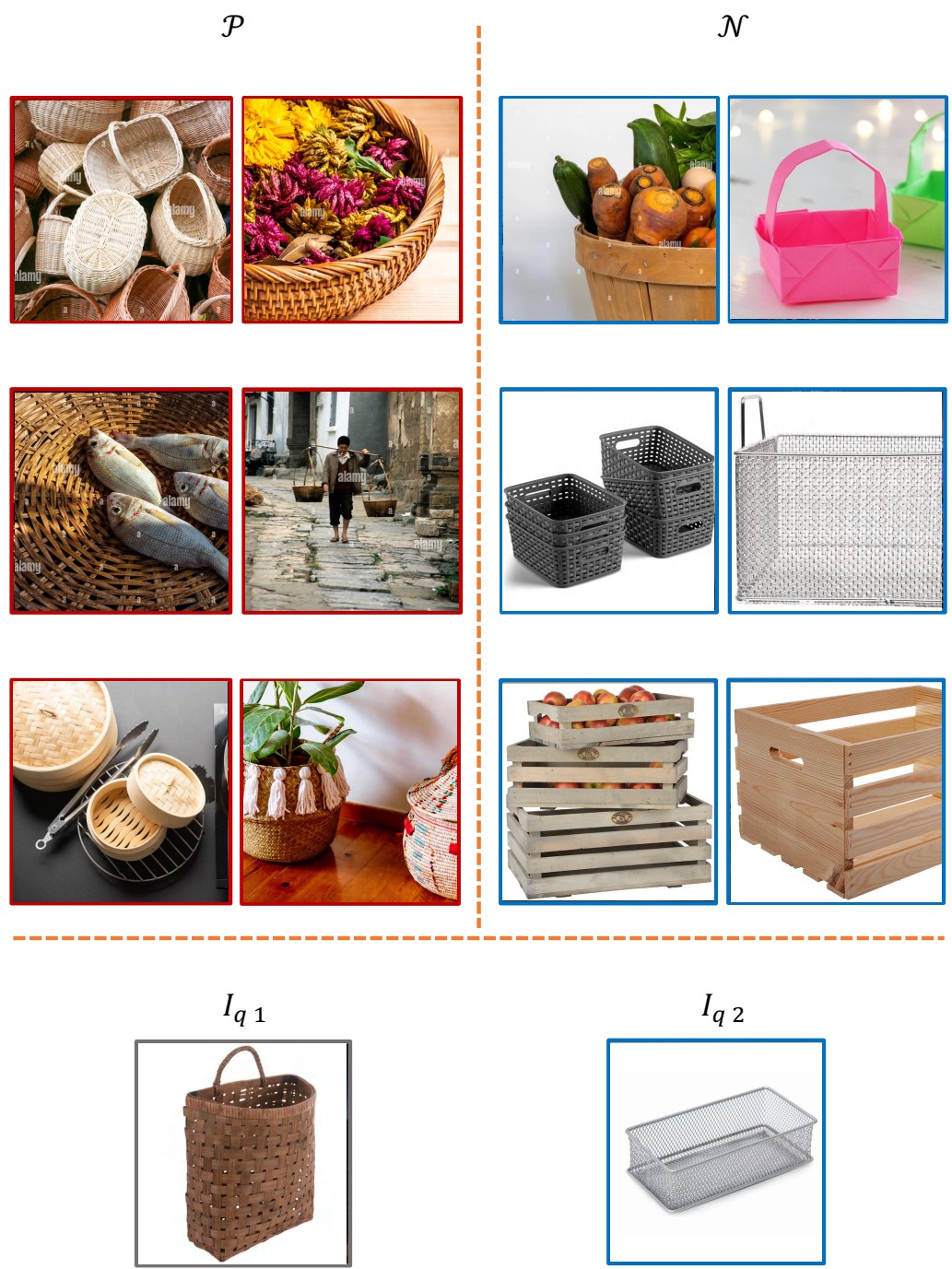

Figure 16: **Example of Bongard-OpenWorld.** The images in set $\mathcal{P}$ all depict *"bamboo baskets"*, while the images in set $\mathcal{N}$ have *"plastic baskets"*, *"metal baskets"* or others, so the concept $\mathcal{C}$ exclusively depicted by $\mathcal{P}$ is *"bamboo baskets"*.

