# OpenReview forum: "Bongard-OpenWorld: Few-Shot Reasoning for Free-form Visual Concepts in the Real World"
_ICLR.cc/2024/Conference — ICLR 2024 poster_

### Official Review · Reviewer_Gtsv · 2023-10-21

**Soundness:** 3 good
**Presentation:** 3 good
**Contribution:** 3 good
**Rating:** 8
**Confidence:** 3

**Summary:**

The paper introduces Bongard-OpenWorld, a benchmark designed to evaluate a system's real-world few-shot reasoning abilities. By incorporating open-world concepts and real images into the classical Bongard Problems, this benchmark serves as a litmus test for current limitations in visual intelligence, motivating further research toward enhancing few-shot reasoning in visual agents. The paper conducts a comprehensive assessment, examining the effectiveness of various Vision-Language Models (VLMs) and Large Language Models (LLMs), as well as proposing a neuro-symbolic reasoning approach tailored for this benchmark.

**Strengths:**

Overall the paper is well written and provides the reader with good insight on why the availability of the proposed benchmark is good for the community as it promotes research into the few-shot reasoning capabilities of current black box deep learning models. The paper presents a tough benchmark that tests current systems on their ability to reason about free-form concepts in a few-shot manner by identifying the common concept in a positive set and distinguishing it from the negative set.

The paper introduces a robust benchmark that assesses systems' ability to perform few-shot reasoning on free-form concepts. It challenges models to identify commonalities in positive sets while distinguishing them from negative sets, enhanced by the inclusion of distractors and hard negatives in the dataset curation process. The evaluation framework covers a wide spectrum, encompassing four distinct approaches: few-shot learning, combined LLM+VLM in single and multiple steps, and a novel neuro-symbolic architecture.

The evaluation setup utilized in the paper is comprehensive and includes the evaluation of four different kinds of approaches that include a few shot learning approaches, LLM+VLM in a single step, LLM+VLM in multiple iteration steps, and finally a proposed neuro-symbolic architecture.

**Weaknesses:**

I would like to look at more variants of the neurosymbolic approach proposed in this work. One avenue worth exploring is a line of research that leverages domain knowledge, such as knowledge graphs, to identify pertinent concepts within an input. Active nodes within the graph could then be employed to pinpoint the common concept within the positive set of images.

The evaluations used in this paper though really comprehensive, miss out on some more ways of evaluation. VLM-based approaches, like GPT4(V), that directly take images as input and can be prompted to obtain the desired input, could be used to identify the relevant concept from a collage of images given together. Since current VLM/LLM approaches are very susceptible to the way they are prompted, it is very important to prompt engineer them in a number of ways and then identify the best working one.

**Questions:**

Table 2 provides a good overview of the performance of various approaches on the proposed benchmark. I would like to see more explanation of the reasoning behind the performance of these approaches. Like for example, Flamingo/ChatGPT/Otter performs significantly worse than the few-shot learning approach SNAIL despite Flamingo/Otter using the same image encoder.

Including a section on failure case analysis for different approaches would be instrumental for the readers in identifying specific challenges and guiding improvements for tackling them.

---

> ### Author Response · Authors · 2023-11-19
> **Response to Reviewer Gtsv (1/N)**
>
> We sincerely thank you for your time and constructive comments. Below, we provide detailed replies to your comments and hope we can resolve your major concerns.
>
> > - I would like to look at more variants of the neurosymbolic approach proposed in this work. One avenue worth exploring is a line of research that leverages domain knowledge, such as knowledge graphs, to identify pertinent concepts within an input. Active nodes within the graph could then be employed to pinpoint the common concept within the positive set of images.
>
> We thank you for the suggestion! Indeed, leveraging KG is a brilliant idea of enhancing the neurosymbolic approach presented in this paper, which currently still replies on perception from other visual foundation model like BLIP. Please feel free to list a few papers that are relevant to the idea you mentioned above, we would love to cite and discuss them in the final version of our paper and if possible, benchmarking them on our dataset. Again, thanks for the insight.
>
>
> > - The evaluations used in this paper though really comprehensive, miss out on some more ways of evaluation. VLM-based approaches, like GPT4(V), that directly take images as input and can be prompted to obtain the desired input, could be used to identify the relevant concept from a collage of images given together. Since current VLM/LLM approaches are very susceptible to the way they are prompted, it is very important to prompt engineer them in a number of ways and then identify the best working one.
>
> Your insights are notably forward-thinking, and we have already prepared results using GPT-4V. All parameters are aligned with GPT-4, except that the called model is *"gpt-4-vision-preview"* and the parameter "detail" set to "high" for fine-grained image representations. We splice raw images together in a manner similar to the examples in Appendix H, using only one query image each time, which is then inputted for GPT-4V reasoning. Here are the results (and they have been merged to the updated version as well). It can be seens that Bongard-OpenWorld remains a great challenge even with GPT-4V. We commit to exploring this further as our prompt engineering might not be that comprehensive given the limited time (and the non-stable performance of their API as well!).
>
> | method | image representation |  short concept | long concept | CS∗ concept | non-CS∗ concept | avg. |
> | :---: | :---: | :---: | :---: | :---: | :---: | :---: |
> | OpenFlamingo | OpenCLIP | 50.0 | 48.4 | 50.9 | 48.6 | 49.3 |
> | Otter | OpenCLIP | 49.3 | 49.3 | 48.9 | 49.4 | 49.3 |
> | GPT-4 | InstructBLIP | 67.3 | 59.7 | 59.3 | 65.6 | 63.8 |
> | GPT-4V | Raw Images | 54.6 | 53.3 | 50.9 | 55.2 | 54.0 |

---

> > ### Author Response · Authors · 2023-11-19
> > **Response to Reviewer Gtsv (2/N)**
> >
> > > - Table 2 provides a good overview of the performance of various approaches on the proposed benchmark. I would like to see more explanation of the reasoning behind the performance of these approaches. Like for example, Flamingo/ChatGPT/Otter performs significantly worse than the few-shot learning approach SNAIL despite Flamingo/Otter using the same image encoder.
> >
> >
> > We thank you for the comments. We woule like to provide some analysis here regarding this issue.
> >
> > First of all, being able to handle open-world concepts like Flamingo/ChatGPT/Otter does not mean the model can solve Bongard-OpenWorld. Because there is one thing that could still be missing in these models: **reasoning-aware perception and holistic perception-reasoning**:
> >
> > - **Separately done perception and reasoning cannot beat Bongard-OpenWorld, no matter how powerful the models are.**
> >
> >     Our benchmark requires extracting common visual concepts globally, rather than extracting atomic concepts from each image separately. The idea is, images in Bongard-OpenWorld have rich details, and therefore it is almost impossible to convert everything the images depicted into text and let LLM or other logic reasoner analyze what is being shared by positives and not depicted by negatives. Instead, VLM-based captioners usually fail to capture the image contents relevant to the current Bongard Problems (see Figure 5 in Appendix G, VLM is distracted by the cardinal and therefore did not describe the concept that is useful for reasoning, which is "tree or branch covered in snow"), therefore leading the follow-up reasoning with LLM fail.
> >
> >
> > - **Holistic perception-reasoning is hard, and it is true for largely pretrained VLMs (as for autumn 2023).**
> >
> >     An ideal model for Bongard-OpenWorld will need to compare images from both positives and negatives, then figure out what is the right concept for classifying query image, i.e. reasoning-aware perception or holistic perception and reasoning. We, therefore, try our best to find an open-sourced VLM that has claimed such capability. Please note that the basic requirement will be handling multiple images at the same time, which already rules out most of the publically available VLMs. We ends up trying OpenFlamingo, Otter and GPT-4V. **Clearly, even the latest and most powerful GPT-4V failed to close the human-machine gap as well.**
> >
> > The above reasons indeed highlight what might be going wrong with ChatGPT, which relies on a seperate VLM captioner, and Flamingo/Otter/GPT-4V, which could fall short on perform the reasoning required by Bongard-OpenWorld. On the contrary, all the few-shot learning baselines (from ProtoNet to SNAIL) have been trained on the training set of Bongard-OpenWorld, therefore they have learned (not perfect tho) to perform the reasoning needed here, while also equipped with open-world or open-vocabulary perception thanks to the OpenCLIP backbone. Therefore, we can see the gap between the best FSL method and these zero-shot large models.
> >
> > We've included these discussions into Appendix B of the revised paper.
> >
> > > - Including a section on failure case analysis for different approaches would be instrumental for the readers in identifying specific challenges and guiding improvements for tackling them.
> >
> > Your suggestions are much appreciated, and we have included more qualitative cases in Appendix G to illustrate these.
> >
> > We hope the above response can resolve your questions and concerns. Please let us know if there is any further question!

---

> > > ### Comment · Reviewer_Gtsv · 2023-11-19
> > > **Response to rebuttal**
> > >
> > > Thank you for the consideration of my comments.
> > >
> > > I appreciate the extensive experimentation conducted with the latest VLM models (GPT4V), the results clearly outline the competitiveness of the proposed benchmarks. The additional sections for the broader impact statement and failure case analysis in the Appendix are also useful for the readers. In terms of the neuro-symbolic approaches that utilize structured knowledge to extract relevant visual concepts from the scene, you may want to look at the papers mentioned below.
> > >
> > > [1] K. Marino, R. Salakhutdinov and A. Gupta, "The More You Know: Using Knowledge Graphs for Image Classification," 2017 IEEE Conference on Computer Vision and Pattern Recognition (CVPR), Honolulu, HI, USA, 2017, pp. 20-28, doi: 10.1109/CVPR.2017.10.
> > >
> > > [2] Bhagat, S., Stepputtis, S., Campbell, J., & Sycara, K.P. (2023). Sample-Efficient Learning of Novel Visual Concepts. ArXiv, abs/2306.09482.

---

> > > > ### Author Response · Authors · 2023-11-20
> > > > **Thank you for your prompt reply**
> > > >
> > > > Dear reviewer,
> > > >
> > > > We thank you for your prompt reply and the recognition of our rebuttal. We're looking into these papers and will try them out. Discussions will be included in the final version as well.
> > > >
> > > > Best,
> > > > Authors

---

### Official Review · Reviewer_CSkd · 2023-10-29

**Soundness:** 3 good
**Presentation:** 3 good
**Contribution:** 3 good
**Rating:** 8
**Confidence:** 4

**Summary:**

The paper introduces a new benchmark called Bongard-OpenWorld, which focuses on open-world visual concept understanding. The task is to classify a query image to belong to one of two sets of images. The positive set contains 6 images depicting a common concept C, such as "animals are running". The negative set contains 6 images of similar concepts but not exactly matching C, e.g., showing a standing animal, or a running robot. The difficulty of this benchmark comes from the two sets sharing common objects or semantics, such that nuances in the semantic concepts need to be understood to perform well. The authors also evaluate relevant existing methods and show that there is still a large gap between current methods and human performance.

**Strengths:**

- The new benchmark and problem setting tackles an important shortcoming of current methods to understand fine-grained semantic concepts in contrast to hard negatives.
- An extensive set of existing methods have been evaluated showing that the best models (64%) are still far from human performance (91%). This makes it a challenging setting for new methods to be developed in the future.
- Evaluations include a few-shot and zero-shot setting, and several different approaches to combine vision-language models and large language models to solve the task.

**Weaknesses:**

- The presentation and writing could be clearer, making some parts of the paper difficult to understand, especially around the creation of the dataset in Sec. 2 and the precise problem setting. For example, it was not immediately clear that the labels of the two sets (positive, negative) are given and do not need to be inferred. The query image can belong to either, but the name "positive" suggests that this is the GT set of the query image, which is not the case. I am adding more clarifying questions in the questions section below.
- The problem setting is imbalanced. The positive set corresponds to a single concept while the negative set does not, but instead contains a subset of the complete concept. While this is not necessarily an issue and can be a design choice, there is no justification why this choice has been made. For instance, why not have both sets correspond to a single concept where the contrasting sets are close in semantics to make it a hard problem? Similarly to how different splits are evaluated in Table 2, it would have helped to show the performance of positive query images vs. negative query images in order to understand if this imbalance makes positive/negative queries easier/harder.
- While a lot of models have been evaluated on the proposed benchmark, a natural baseline is missing: computing the image similarity between the query image and the two sets. For instance, one can use any pre-trained image encoder (CLIP, Dino, etc.) or image-to-image retrieval method and use the mean similarly of the image embeddings per set to make a prediction. Using captioning models and LLMs seems to introduce complexity while at the same time discarding fine-grained image information by only relying on text to make the decision.
- With around 1K tasks it is a rather small dataset. Hence, focusing on the "zero-shot" setting without involving training might be the better use case.
- While it is true that the benchmark contains a large variety of concepts, positioning it heavily as an "open-world" and "open-vocabulary" task could be a bit misleading as the core problem is to identify whether an image came from set A or set B. The optional task of naming the concept is most fitting for "open-world", but it serves a minor role in the paper.

**Questions:**

- Have you thought about not providing the labels "positive" and "negative" for the two sets to the methods? Why have you chosen this setup?
- In Sec 2.1: What is a grid in this context? What is grid sampling? How do you define "concept tuples"? Neither the main paper, nor the supplementary clarifies this sufficiently.
- How do you ensure that the dataset does not contain duplicate concepts? I assume this is the case because in Table 1, it is reported that the dataset has 1.01K concepts and 1.01K tasks.
- What are the exact instructions the annotators were given? For instance, when "annotators are instructed to write visual concepts by following a predefined set of categories illustrated in Table 7" and when "they are also asked to combine these challenging concepts with those mined from CC-3M" (Sec. 2.1).
- Images are collected by using an online search based on the concepts. What is the license of the images collected? Do the authors have the rights to distribute the images?
- In Sec. 2.2. you write: "the annotators are then asked to provide two sets of candidates for positive and negative queries". How many images are collected here for possible query images? Why choose only one positive and negative image as query in the end?
- Does defining the concepts of category 0 (from CC3M) undergo any crowd-sourcing or is it fully automated?
- Is there performance difference between positive and negative queries?
- In Figure 2c, both x and y-axis should be labeled. What is the scale/size of the number of concepts (x-axis)? What is the unit of the numbers on the y-axis?
- What is meant by "we report the overall accuracy of all models". Does Table 2 report test set accuracy or accuracy over the whole dataset, i.e., including training samples?
- Why are concepts from CC3M considered non-commonsense?
- How is ChatGPT finetuned (Table 2)? Does this use the finetuning API of OpenAI? More details would help make this more reproducible.

Comments/suggestions:
- Table 2 includes methods that use training data to update NN weights and others that do not update weights ("zero-shot" setting). It would be much clearer if the table indicates which models use training data.
- The following phrase appears 3 times in the manuscript. I suggest to to reduce this repetition and rephrase it according to the context. "We even designed a neuro-symbolic reasoning approach that reconciles LLMs & VLMs with logical reasoning to emulate the human problem-solving process for Bongard problems".
- Sec. 3.2 (at the end) promises captioning metrics, but they do not appear in the main paper, only in the supplementary.
- The formatting of Table 7 is confusing. It would be better to clearly separate the left half from the right half, or simply just make it 10 rows.

---

> ### Author Response · Authors · 2023-11-19
> **Response to Reviewer CSkd (1/N)**
>
> We sincerely thank you for your time and constructive comments. Below, we provide detailed replies to your comments and hope we can resolve your major concerns.
>
>
> > - The presentation and writing could be clearer, making some parts of the paper difficult to understand, especially around the creation of the dataset in Sec. 2 and the precise problem setting. For example, it was not immediately clear that the labels of the two sets (positive, negative) are given and do not need to be inferred. The query image can belong to either, but the name "positive" suggests that this is the GT set of the query image, which is not the case. I am adding more clarifying questions in the questions section below.
>
> Thanks for highlighting this point, we have revised our main paper for greater clarity. For your specific question here, the labels of the support set are given and without anything else. Each Bongard Problem includes both a positive and a negative query image.
>
> > - The problem setting is imbalanced. The positive set corresponds to a single concept while the negative set does not, but instead contains a subset of the complete concept. While this is not necessarily an issue and can be a design choice, there is no justification why this choice has been made. For instance, why not have both sets correspond to a single concept where the contrasting sets are close in semantics to make it a hard problem?
>
> Our problem setting adheres strictly to the original Bongard Problem (a classic puzzle of computer vision and pattern recognition: https://en.wikipedia.org/wiki/Bongard_problem), which include several features:
>
> - The support set comprises only images of positive and negative with their labels (just indicates whether it belongs to the positive or negative set).
> - Images from the positive set should depict the same concept, while those belong to the negative set should not depict this concept and the concept depicted by the negative images can be varied -- this could lead to an "imbalance" on the number of concepts but yes, this is a designed choice made by the original Bongard problem.
>
> But Bongard-OpenWorld also introduces some new features:
>
> - First of all, both the positive and negative images includes distractor concepts (as we've demonstrated in Figure 1 of the original paper) -- that is, images from the positive set are in fact could also depict many more concepts besides the ground truth, alleviating the "imbalance" issue.
>
> - Secondly, as you can see from the example problems we showed in the Appendix H, when we picking the negative images, we also seek to make them share some common concepts, this is relevant to the solution you mentioned above and indeed help create a hard problem.
>
> - Finally, we even make positive and negative images could overlap on the concepts they depict. For example, given the ground truth concept is "red tie", both positie and negative concepts can depict "dog", resulting "dog wearing a red tie" (positive) and "dog wearing a blue shirt" (negative).
>
> Again, we thank you for raising this issue, and we believe our design in Bongard-OpenWorld not only respect the original Bongard problem setting, but also mitigate the possible drawbacks, including imblancing and easy problems.
>
> > - ...Similarly to how different splits are evaluated in Table 2, it would have helped to show the performance of positive query images vs. negative query images in order to understand if this imbalance makes positive/negative queries easier/harder.
> > - Is there performance difference between positive and negative queries?
>
> We thank you for your suggestion!  Here are some additional results singling out the accuracy on positive and negative queries. As you can see, there might be slight difference for a single model, but we did not observe a pattern or preference towards positive or negative queries globally. Therefore Bongard-OpenWorld does not appear to exhibit a strong imbalance issue.
>
> | method | image representation | aux. task? | positive query | negative query | avg. |
> | :---: | :---: | :---: | :---: | :---: | :---: |
> | Meta-Baseline | OpenCLIP | &#10004; | 59.5 | 53.0 | 56.3 |
> | MetaOptNet | OpenCLIP | &#10004; | 57.9 | 57.0 | 57.5 |
> | ProtoNet | OpenCLIP | &#10004; | 58.2 | 58.7 | 58.5 |
> | SNAIL | OpenCLIP | &#10004; | 61.2 | 66.8 | 64.0 |
> | GPT-4 | InstructBLIP | &#10004; | 67.4 | 60.2 | 63.8 |
>
> > - Have you thought about not providing the labels "positive" and "negative" for the two sets to the methods? Why have you chosen this setup?
>
> As we mentioned above, not proving the positive or negative labels could make the setting largely deviate from the original Bongard problem setting, which we choose to respect when creating this dataset. Also, we don't see a clear advantages of doing such. That include the "imbalancing" issue in your comment -- our response above has shown that our dataset does not appear to have a strong imbalance issue.

---

> > ### Author Response · Authors · 2023-11-19
> > **Response to Reviewer CSkd (2/N)**
> >
> > > - In Sec. 2.2. you write: "the annotators are then asked to provide two sets of candidates for positive and negative queries". How many images are collected here for possible query images? Why choose only one positive and negative image as query in the end?
> >
> > Sorry for not making it clear. We start with 7 query images for both positive and negative and rule it down to 1. The choice of only using 1 query is to align with the original Bongard problem, which we respect in Bongard-OpenWorld.
> >
> > > - While a lot of models have been evaluated on the proposed benchmark, a natural baseline is missing: computing the image similarity between the query image and the two sets. For instance, one can use any pre-trained image encoder (CLIP, Dino, etc.) or image-to-image retrieval method and use the mean similarly of the image embeddings per set to make a prediction. Using captioning models and LLMs seems to introduce complexity while at the same time discarding fine-grained image information by only relying on text to make the decision.
> >
> > Thanks for your insight. We provide the results of the natural baseline you mentioned below. In fact, this baseline could be very similar to ProtoNet (which we already benchmarked), but ProtoNet further fine-tunes the visual features on the training set.
> >
> > This is how we implement it: we calculates the mean similarity between a query image embedding (produced by CLIP) and both positive and negative images embeddings (also produced by CLIP) in the support set, assigning the label of the higher mean as its prediction.
> >
> > As for the result, its notable underperformance compared to coin-flipping (50% chance of success) stems from adversarial query selection (section 2.2). In selecting query images, we deliberately increase the similarity (measured by CLIP embedding distance) between positive queries and negative images in order to make the problem harder. Such design adversarally prevent this natural baseline with CLIP image embeddings from performing well.
> >
> > Please note that, even with other backbone (ex. DINO), we believe the problem should still exist, as this natural baseline can be closely resemble to ProtoNet, which also fine-tunes the backbone. In our experiment, we already demonstrate its inability on Bongard-OpenWorld, even with non-CLIP backbone.
> >
> > | method | image representation | aux. task? | short concept | long concept | CS∗ concept | non-CS∗ concept | avg. |
> > | :---: | :---: | :---: | :---: | :---: | :---: | :---: | :---: |
> > | natural baseline | OpenCLIP | &#10007; | 12.8 | 7.7 | 12.7 | 9.7 | 10.5 |
> > | ProtoNet | OpenCLIP | &#10004; | 59.2 | 57.7 | 51.8 | 61.0 | 58.5 |
> >
> > > - With around 1K tasks it is a rather small dataset. Hence, focusing on the "zero-shot" setting without involving training might be the better use case.
> >
> > Thanks for your insight. Our evaluation includes VLM, LLM, and neuro-symbolic approaches, and all of them exclusively utilize the test set, primarily on zero-shot settings. However, we have also explored diverse few-shot learning approaches to ensure a more comprehensive evaluation in our benchmark.
> >
> > > - While it is true that the benchmark contains a large variety of concepts, positioning it heavily as an "open-world" and "open-vocabulary" task could be a bit misleading as the core problem is to identify whether an image came from set A or set B. The optional task of naming the concept is most fitting for "open-world", but it serves a minor role in the paper.
> >
> > The core of "open-world" and "open-vocabulary" lies in its thinking process of solving the Bongard Problems, instead of merely examining its output. That is, to solve Bongard problem, namely, predicting whether an image came from set A or set B, the model has to understand and reason these images in an open-world or open vocabulary fashion. Therefore, we believe the position we choose is appropriate.
> >
> > > - In Sec 2.1: What is a grid in this context? What is grid sampling? How do you define "concept tuples"? Neither the main paper, nor the supplementary clarifies this sufficiently.
> >
> > We apologize for this oversight and have updated our main paper. Regarding your specific inquiries here, 'grid' denotes the pool we sample concept from. Instead of sampling from the whole CC-3M concept set, we first split it into seveal small pools, or as we name it, "grids", with size of 300 each. Then we perform top-k sampling within each grid to obtain concepts. We find this balances the need for sampling top concepts and sample diversity.
> >
> > Furthermore, 'concept tuples' refer to candidates for the visual concepts $\mathcal{C}$, where each concept conprises a tuple of words, ex. "<red, tie>".

---

> > > ### Author Response · Authors · 2023-11-19
> > > **Response to Reviewer CSkd (3/N)**
> > >
> > > > - How do you ensure that the dataset does not contain duplicate concepts? I assume this is the case because in Table 1, it is reported that the dataset has 1.01K concepts and 1.01K tasks.
> > >
> > > Each Bongard Problem in our proposed benchmark has undergone extensive manual review to ensure full compliance with standard settings and is unique, even if filtered during automated collection. Thanks for your attention to this matter.
> > >
> > > > - What are the exact instructions the annotators were given? For instance, when "annotators are instructed to write visual concepts by following a predefined set of categories illustrated in Table 7" and when "they are also asked to combine these challenging concepts with those mined from CC-3M" (Sec. 2.1).
> > >
> > > Here are the guides to our annotators:
> > >
> > > - General concept annotation guide:
> > >     1. Positive images all share one concrete concept, denoted as $\mathcal{C}$. This concept should be engaging and not overly simplistic (such as basic objects, 'dog' or 'cat'), but rather somewhat abstract (for instance, 'running' or 'reflection').
> > >     2. Each negative image must depict a concept that must have some added distractors with $\mathcal{C}$.
> > >     3. Each negative image must depict a concept that partially overlaps with $\mathcal{C}$.
> > >
> > > - Commonsense annotation guide:
> > >
> > >     Follow the definition of commonsense, manually revise the automatically collected candidate concepts (from CC-3M) to make them more "commonsense"-alike.
> > >
> > >     One definition of commonsense from Wikipedia: "Commonsense knowledge includes the basic facts about events (including actions) and their effects, facts about knowledge and how it is obtained, facts about beliefs and desires. It also includes the basic facts about material objects and their properties." The following are some examples:
> > >
> > >     1. HOI: discerning subtle interaction between humans and objects, e.g. playing basketball vs. playing volleyball.
> > >     2. Taste / Nutrition / Food: identifying ingredients in food, e.g. high-calorie vs. high-sugar.
> > >     3. Color / Material / Shape: differentiating similar materials with various appearances, e.g. rough wood vs. painted wood.
> > >     4. Functionality / Status / Affordance: recognizing variants that have the same function, e.g. cuttable.
> > >     5. And / Or / Not: making logical judgments about something, e.g. no something.
> > >     6. Factual Knowledge: possessing factual knowledge, e.g. buildings in the capital of the United States.
> > >     7. Meta Class: induction of metaclass, e.g. rescue vehicle (including ambulance, police car, fire engine, etc), Felidae (including cat, lion, puma, etc).
> > >     8. Relationship: perceiving spatial relationships, e.g. "near", which may solely be "near on pixel distance" but located in the foreground and background respectively results in "far on physical distance".
> > >     9. Unusual observations: e.g. light refraction, gaze direction.
> > >
> > > As you can see, the annotators are guided to revise & reorganize the original CC-3M concepts into commonsense concepts, with explicit guidance on what can be viewed as commonsense concepts. This aligns with the descriptions in the main paper. We've included this guide in the Appendix C to make it more clear.
> > >
> > > > - Why are concepts from CC3M considered non-commonsense?
> > >
> > > Thanks for pointing this out, per our observation (as we mentioned above, each Bongard problems will be manually screened), most concepts mined from CC-3M and used by Bongard-OpenWorld indeed do not exhibit commonsense characteristic as we showed in the previous response. But indeed, we've removed this claim, as the commonsense category is labelled after the construction of Bongard problems, which means both problems from human annotated commonsense concepts or the mined CC-3M concepts can be labelled as commonsense problems.
> > >
> > > > - Images are collected by using an online search based on the concepts. What is the license of the images collected? Do the authors have the rights to distribute the images?
> > >
> > > We do not set further restrictions on searching therefore the license could be varied; by default, we only distribute URLs instead of the original image (we do have cache of the original images as backup); thanks for pointing this out, we are contacting the copyright owner about the distribution issue.
> > >
> > > > - Does defining the concepts of category 0 (from CC3M) undergo any crowd-sourcing or is it fully automated?
> > >
> > > All the data collection processes in our proposed benchmark combine automated preparation of candidates with meticulous manual review to ensure complete compliance with our established criteria.

---

> > > > ### Author Response · Authors · 2023-11-19
> > > > **Response to Reviewer CSkd (4/N)**
> > > >
> > > > > - In Figure 2c, both x and y-axis should be labeled. What is the scale/size of the number of concepts (x-axis)? What is the unit of the numbers on the y-axis?
> > > >
> > > > In Figure 2c, x represents an integer sequence ranging from 0 to the total count of unique word (approximately 1.6K) that could appear in the concepts of Bongard-OpenWorld. Meanwhile, y denotes the ratio of each word across all concepts (1.01K), 'people' as the most frequent word, with its ratio being approximately eight in a thousand.
> > > >
> > > > > - What is meant by "we report the overall accuracy of all models". Does Table 2 report test set accuracy or accuracy over the whole dataset, i.e., including training samples?
> > > >
> > > > All accuracy presented in Table 2 were derived from the test set only.
> > > >
> > > > > - How is ChatGPT finetuned (Table 2)? Does this use the finetuning API of OpenAI? More details would help make this more reproducible.
> > > >
> > > > We used the fine tuning API of OpenAI and the training set of our proposed benchmark. To ensure complete reproducibility, in our abstract, we have clearly stated that all the code implementation details and the dataset have been made public under this anonymous github repo: https://github.com/Bongard-OpenWorld/Bongard-OpenWorld.
> > > >
> > > > > - Table 2 includes methods that use training data to update NN weights and others that do not update weights ("zero-shot" setting). It would be much clearer if the table indicates which models use training data.
> > > >
> > > > We have updated Table 2, thanks for your reminder.
> > > >
> > > > > - The following phrase appears 3 times in the manuscript. I suggest to to reduce this repetition and rephrase it according to the context. "We even designed a neuro-symbolic reasoning approach that reconciles LLMs & VLMs with logical reasoning to emulate the human problem-solving process for Bongard problems".
> > > >
> > > > We sincerely apologize for our oversight and have rewritten these sentences. Your meticulous attention is greatly appreciated.
> > > >
> > > > > - Sec. 3.2 (at the end) promises captioning metrics, but they do not appear in the main paper, only in the supplementary.
> > > >
> > > > Owing to space limits, we have relocated this to the Appendix and accordingly restructured the article. These metrics are conventional for caption evaluation; references for these are provided in Appendix C.
> > > >
> > > > > - The formatting of Table 7 is confusing. It would be better to clearly separate the left half from the right half, or simply just make it 10 rows.
> > > >
> > > > Thanks for your suggestion, we have split this Table.
> > > >
> > > >
> > > > We hope the above response can resolve your questions and concerns. Please let us know if there is any further question!

---

> > > > > ### Comment · Reviewer_CSkd · 2023-11-20
> > > > >
> > > > > I would like to thank the authors for their comprehensive response addressing my questions. Many changes have already improved the manuscript. Below, I am sharing follow-up comments on some of the responses.
> > > > >
> > > > > Several of my questions have been answered by stating that some design choices of the proposed setting have been made in order to adhere to the original Bongard problem formulation. These questions include:
> > > > > > [...] For instance, why not have both sets correspond to a single concept where the contrasting sets are close in semantics to make it a hard problem?
> > > > >
> > > > > > Our problem setting adheres strictly to the original Bongard Problem (a classic puzzle of computer vision and pattern recognition: https://en.wikipedia.org/wiki/Bongard_problem), which include several features:
> > > > > > - [...]
> > > > > > - Images from the positive set should depict the same concept, while those belong to the negative set should not depict this concept and the concept depicted by the negative images can be varied -- this could lead to an "imbalance" on the number of concepts but yes, this is a designed choice made by the original Bongard problem.
> > > > > ---
> > > > > > Have you thought about not providing the labels "positive" and "negative" for the two sets to the methods? Why have you chosen this setup?
> > > > >
> > > > > > As we mentioned above, not proving the positive or negative labels could make the setting largely deviate from the original Bongard problem setting, which we choose to respect when creating this dataset. [...]
> > > > > ---
> > > > > > [...] Why choose only one positive and negative image as query in the end?
> > > > >
> > > > > > [...] The choice of only using 1 query is to align with the original Bongard problem, which we respect in Bongard-OpenWorld.
> > > > >
> > > > > Regardless of how the original Bongard problem was formulated, I think important design choices of contemporary benchmarks should be justified not only because the challenges of computer vision change over time, but also because it is important to understand what we are measuring when evaluating models on new benchmarks.
> > > > >
> > > > > To further make a point, I don't think Bongard-OpenWorld actually strictly follows the original setting of Bongard problems, which, by the way, might not be well defined by the linked Wikipedia article. For instance, [https://www.oebp.org/welcome.php](https://www.oebp.org/welcome.php) describe the problem as:
> > > > > > A Bongard Problem traditionally includes
> > > > > > - a dividing line,
> > > > > > - six images on the left side of the dividing line, and
> > > > > > - six images on the right side of the dividing line.
> > > > > > Most important, there is some simple description that fits all the images on the left side (but none on the right) and, oppositely, a simple description that fits all the images on the right side (but none on the left).
> > > > >
> > > > > And in M. M. Bongard's "Pattern recognition" from which these problems originate, they follow the above description and there is always an simple common description for all images on either side. For example, the solutions of the first 3 problems from Appendix 3 are
> > > > > > Empty picture <-> Not empty picture
> > > > > > Large figures <-> Small figures
> > > > > > Outline figures <-> Solid figures
> > > > >
> > > > > So I would argue, originally the two sets are symmetrical, not labeled (positive/negative), and there is actually no "query" image, but the task is to simply name the common concept on either side.
> > > > >
> > > > > My point is: It is fine to adhere or deviate from a previously established setting, but there should still be a proper reason behind each choice such that the benchmark targets a clear goal in computer vision research. For some of my original questions above, good reasoning is currently missing.
> > > > >
> > > > > >  Here are some additional results singling out the accuracy on positive and negative queries.
> > > > >
> > > > > I think these results should be included in the paper (at least in the appendix).
> > > > >
> > > > > > Please note that, even with other backbone (ex. DINO), we believe the problem should still exist, as this natural baseline can be closely resemble to ProtoNet, which also fine-tunes the backbone. In our experiment, we already demonstrate its inability on Bongard-OpenWorld, even with non-CLIP backbone.
> > > > >
> > > > > While I understand the motivation behind adversarial query selection was to make the benchmark challenging, it might also introduce this rather unnatural behavior and penalty for reasonable approaches that use contrastively trained VLMs (such as CLIP) and any models that build on top of pre-trained CLIP models. I think testing a separate baseline (e.g. DINO) could help better understand this dynamic. For full transparency, I suggest including the "natural baseline" into the paper.
> > > > >
> > > > > > [...] both problems from human annotated commonsense concepts or the mined CC-3M concepts can be labelled as commonsense problems.
> > > > >
> > > > > I agree with this sentiment. From my understanding, there are still some references where I suggest to rename "non-CS": Table 1, Figure 2, Table 7, Sec. 2.3.

---

> ### Author Response · Authors · 2023-11-21
> **Thank you for the promot reply! We really appreciate it.**
>
> Dear Reviewer CSkd,
>
> We would like to thank you for your prompt reply and we're happy to see some of your concerns have been addressed. Per the remaining issues, we would like to offer some further clarifications:
>
> > My point is: It is fine to adhere or deviate from a previously established setting, but there should still be a proper reason behind each choice such that the benchmark targets a clear goal in computer vision research. For some of my original questions above, good reasoning is currently missing.
>
> We thank you for posting the references to the original Bongard problem. Indeed, our implementation slightly deviates from the original Bongard problem, in terms of the following:
>
> - The two sets are not symmetrical -- in our case, only set A depicts a shared concept, while set B does not
>
> - We add a query image and switch the main task from recognizing the concept (we still have it, as the captioning task, but it is not the "main task" that is presented in the "main result" section of the paper) to determine whether the query belongs to set A or set B.
>
> - Some additional features on image selection, as we mentioned in the response above.
>
> Now we dive into some reasoning on the various design choices (why making these modifications, why introducing several additional features, etc), including what we would like to benchmark, and how is this important to computer vision research.
>
> 1. both sets with labels, also an additional query
>
>     To be more accurate, this "non-symmetrical" setting is effectively inherited from two seminal works: Bongard-LOGO[1] and Bongard-HOI[2]. We believe the idea behind this is: **the original Bongard problem is rather challenging**, as it requires the machine to directly produce natural language descriptions of the concept after reading images from both sets. Even with today's powerful VLMs, this is still quite difficult. Therefore, it is natural to simplify the question a bit, let's say, turning the original "concept regression" problem into classification, so more approaches, including many popular few-shot image classification methods (ProtoNet, MetaBaselines, etc) can be benchmarked here.
>     A good idea would be to additionally introduce a query image, and categorize which set it belongs to. This creates a classification problem, while maintaining the core idea of the original Bongard problem -- to identify the concept of a set of images. Evidently, if the query is properly chosen, only when the model is able to identify the true concept and (implicitly) develop an image classifier for this concept, the query can be correctly categorized.
>     From a computer vision research perspective, such modification democratizes the evaluation of the Bongard problem by creating a "simpler" version of it, and makes it possible to examine the progress of current computer vision models in terms of few-shot concept understanding, which is a very interesting yet crucial capability that is originally benchmarked by Bongard problems.
>
> 2. two "non-symmetrical" sets
>
>     This is also inherited from Bongard-LOGO[1] and Bongard-HOI[2]. Compared to the original Bongard problems with two symmetrical sets, creating a non-symmetric "negative" set actually streamlines the reasoning process a bit. In the original Bongard problem, the concept induction process is repeated. We likely start with some hypothesis of the concept being depicted by set A, then we use set B images to refine them by ruling out those being depicted by set B. Next, we move to set B and do the same. If there is no clear answer, we may have to repeat the aforementioned steps until a verdict is returned. It can be seen that the induction processes of concepts from set A and set B are symmetrical as well. Therefore, by making the shared concept only exist in set A, we remove the reasoning process that is symmetrical to this in the original Bongard problem and therefore streamline the overall reasoning process.
>     We admit doing such could introduce some "imbalance" issues. But as we already mentioned in the previous response, we've introduced several techniques to mitigate this.
>     Back to what this design choice means to computer vision research in general: the idea is similar to 1., we would like to simplify the problem a bit so examining the progress of existing models, which are not strong enough to solve the original Bongard problem directly.

---

> > ### Author Response · Authors · 2023-11-21
> >
> > 3. overlapped positive and negative images (in terms of images) and distractions
> >
> >     The idea here is to make the classification problem more difficult. For example, if the set A concept is "dog with a blue tie", while the set B images does not contain any of these, ex. dog, blue, tie, etc, the model will be able to classify the query image simply with a "dog" classifier, which undermine the goal of recognizing the full concept of this problem. Therefore, making the concepts of the images from these two sets overlap a bit (without violating the requirement that images from set B cannot depict the true shared concept of set A), and further, adding distractor concepts, could significantly make the problem more challenging and less possible to have shortcut solutions. From a computer vision research perspective, this could lead to more effective evaluation, as we can make sure the capability being tested on our problem is indeed few-shot learning of free-form visual concepts, not something much simpler.
> >
> > [1] https://arxiv.org/abs/2010.00763
> >
> > [2] https://arxiv.org/abs/2205.13803
> >
> >
> > > I think these results should be included in the paper (at least in the appendix).
> >
> > Just did. Please take a look at **Appendix E.3**.
> >
> > > While I understand the motivation behind adversarial query selection was to make the benchmark challenging, it might also introduce this rather unnatural behavior and penalty for reasonable approaches that use contrastively trained VLMs (such as CLIP) and any models that build on top of pre-trained CLIP models. I think testing a separate baseline (e.g. DINO) could help better understand this dynamic. For full transparency, I suggest including the "natural baseline" into the paper.
> >
> > Here are the results:
> >
> > | method | image representation | aux. task? | short concept | long concept | CS∗ concept | non-CS∗ concept | avg. |
> > | :---: | :---: | :---: | :---: | :---: | :---: | :---: | :---: |
> > | natural baseline | OpenCLIP | &#10007; | 12.8 | 7.7 | 12.7 | 9.7 | 10.5 |
> > | natural baseline | DINO | &#10007; | 18.3 | 11.0 | 16.4 | 14.5 | 15.0 |
> > | natural baseline | DINOv2 | &#10007; | 17.9 | 11.0 | 14.5 | 14.8 | 14.8 |
> > | ProtoNet | OpenCLIP | &#10004; | 59.2 | 57.7 | 51.8 | 61.0 | 58.5 |
> >
> > Note:
> > - The model architecture for [CLIP](https://github.com/mlfoundations/open_clip) is ViT-H/14, which reached a performance of 78.0% on the ImageNet top-1 accuracy.
> > - The model architecture for [DINO](https://github.com/facebookresearch/dino) is ViT-B/8, which achieved a performance of 80.1% on the ImageNet top-1 accuracy.
> > - The model architecture for [DINOv2](https://github.com/facebookresearch/dinov2) is ViT-g/14 (with registers), achieving a performance of 87.1% on the ImageNet top-1 accuracy.
> >
> > It can be seen that, natural baselines with both CLIP and DINO as backbone fail to produce comparable performances to the existing baselines. Our hypothesis aligns with the initial analysis with CLIP -- when constructing Bongard-OpenWorld, we delibrately increase the similarity between positive queries and negative images in order to make the problem harder. Although the similarity is measured with CLIP embedding disantance, as the results above suggested, embeddings based on other approaches, ex. DINO and DINO-v2 could also be affected.
> >
> > We've included these results in the updated paper (**Appendix E.4**).
> >
> > > I agree with this sentiment. From my understanding, there are still some references where I suggest to rename "non-CS": Table 1, Figure 2, Table 7, Sec. 2.3.
> >
> > All fixed. We've replaced all "non-CS" with "anything else". Feel free to let us know if you have better recommendation!
> >
> > Again, thank you for the thoughtful feedback. We're looking forward to hearing from you soon.
> >
> > Best,
> >
> > Authors

---

> > > ### Comment · Reviewer_CSkd · 2023-11-21
> > >
> > > Thank you for providing an answer equally promptly.
> > >
> > > I find your additional arguments on the design choices more convincing then in the previous message. The elaborations are much appreciated. It seems reasonable that the trade-offs for focusing on classification rather than concept identification as well as using a negative set, simplifies the dataset creation and provides an adequate challenge for present models and research.
> > >
> > > I still believe a balanced problem setting, where each set of images correspond to a clear concept (not only negations on a fine-grained level), is more desirable but I now better understand the reason behind this choice.
> > >
> > > > It can be seen that, natural baselines with both CLIP and DINO as backbone fail to produce comparable performances to the existing baselines.
> > >
> > > From my understanding, this is primarily happening because of the adversarial query selection (based on CLIP score) of 1 from 7 possible query images for both positive and negative sets. I am still concerned about this fact because it is unclear whether this benchmarks artificially discourages certain solutions by design which might not reflect a real world setting. Ideally, a benchmark should be neutral in such a way that researchers can find the best solution without being penalized for choosing certain models, such as common image encoders.
> > >
> > > It is related to my initial question about the choice to only select a single query image. (I didn't mean to have multiple queries for a single task, but just extend the dataset with all possible queries.) Alternatively, all 7 image could have served as test/query images.
> > > This not only increases the sample size for the test set, but could also more naturally reflect real world examples including both simpler and more difficult images.
> > >
> > > It seems quite counterintuitive that strong vision models such has CLIP and DINO perform so poorly and way below a random baseline.

---

> > > > ### Author Response · Authors · 2023-11-22
> > > >
> > > > Dear reviewer CSkd,
> > > >
> > > > Thank you for the feedback and we're so glad to see the additional reasoning works out! Below, let us talk you through the remaining issues:
> > > >
> > > > **On whether Bongard-OpenWorld discourages certain models**
> > > >
> > > > We thank you for raising this very important issue. Indeed, our benchmark should be neutral without penalizing certain baselines. However, we would like to point out that, our results so far (including those in the paper, and the additional results in this thread), at least, do not provide evidence of our benchmark being biased to discourage certain backbones. Here are our arguments:
> > > >
> > > > Although our adversarial query selection is based on the CLIP embedding distance, it somehow also generalizes to distance metrics with other backbones as well. Our additional results with natural baseline + DINO/DINOv2 have demonstrated **equally** worse results to natural baseline + CLIP. To further verify this point, here are some results of the natural baseline + Imagenet-22k supervised pretrained model, which delivers **similar** results. Therefore, these results do not provide evidence that our benchmark, which is a result of such adversarial query selection, discourages certain backbone usage.
> > > >
> > > > | method | image representation | aux. task? | short concept | long concept | CS∗ concept | non-CS∗ concept | avg. |
> > > > | :---: | :---: | :---: | :---: | :---: | :---: | :---: | :---: |
> > > > | natural baseline | IN-22K | &#10007; | 15.6 | 8.8 | 10.0 | 13.4 | 12.5 |
> > > > | natural baseline | OpenCLIP | &#10007; | 12.8 | 7.7 | 12.7 | 9.7 | 10.5 |
> > > > | natural baseline | DINO | &#10007; | 18.3 | 11.0 | 16.4 | 14.5 | 15.0 |
> > > > | natural baseline | DINOv2 | &#10007; | 17.9 | 11.0 | 14.5 | 14.8 | 14.8 |
> > > > | ProtoNet | OpenCLIP | &#10004; | 59.2 | 57.7 | 51.8 | 61.0 | 58.5 |
> > > >
> > > > (The model architecture for [IN-22K](https://github.com/facebookresearch/ConvNeXt) is ConvNeXt-B/224, which reached a performance of 85.8% on the ImageNet top-1 accuracy.
> > > > )
> > > >
> > > > We need to point out that, we don't have direct evidence for the opposite -- all backbones are treated equally, either, as finding proof for that could be basically intractable. However, you may take a sip from our main results -- from training from scratch to ImagNet-1k/22k and OpenCLIP, most approaches exhibit steady improvement thanks to the "open-worldness" of visual backbones. If Bongard-OpenWorld is so biased and discourages certain backbones, such biases could neutralize/massage the progress made by more open-world backbones and, likely, we won't be able to observe this trend. None of this actually happened and the trend is clear.
> > > >
> > > > So why the impact caused by a CLIP-based metric could generalize to other backbones, or "It seems quite counterintuitive that strong vision models such as CLIP and DINO perform so poorly and way below a random baseline"? Here are some of our hypotheses:
> > > >
> > > > - CLIP-based metrics ensure the query is close to the opposite set of images (pos query to neg images, neg query to pos images), both visually and semantically, thanks to its superior characteristics. In fact, CLIP can not only provide semantic information, but it can also preserve some visual detail or even geometry information, as suggested by [1], which utilizes a CLIP-based encoder for subject-driven editing tasks, where the subject is encoded by a CLIP encoder. Most of the visual backbones, either capture visual (ex. self-supervised method like DINO) or semantic (supervised methods like ImageNet pretrained) information of the images or both (CLIP). Therefore, a close CLIP distance could mean the two images are both visually and semantically close, and therefore a query selected via this can confuse many backbones, not just CLIP-based backbones.
> > > >
> > > > To really combat this intended confusion made by our adversarial selection, the model must learn to adjust its perception to the images based on the current Bongard problem context, i.e. the visual backbone has to be tuned in a way, or the reasoning part of the model has to be learnable to select certain features provided by the backbone. Merely reasoning with the backbone distance, as the natural baseline does, will therefore fall just into the trap we set.
> > > >
> > > > **On why the query cannot be randomly selected from the set A/B**
> > > >
> > > > As we've mentioned above, our adversarial query selection is essential to the success of Bongard-OpenWorld as we force the model to really reason about the concepts rather than leveraging some superficial features. This becomes truer as we uses real-world images, not synthetic images. And since the query has to be adversarially selected, not all images from the set A/B can become the query. This is indeed a trade-off to be made but we have to choose this route aiming for a more challenging dataset.
> > > >
> > > > Best,
> > > >
> > > > Authors
> > > >
> > > > [1] https://arxiv.org/abs/2305.14720

---

> > > > > ### Comment · Reviewer_CSkd · 2023-11-23
> > > > >
> > > > > I would like to thank the authors for their time and effort in providing additional insights regarding my questions.
> > > > >
> > > > > In general, I follow your argumentation, although I still find it surprising how these models consistently deviate below the random baseline. There seems to be on average higher visual similarity between the opposing queries and sets while maybe a fine detail determines the actual GT concept.
> > > > >
> > > > > In conclusion, the rebuttal has addressed my questions. Preliminary, I am raising my score to 6 and will potentially make further adjustments to my evaluation after the reviewer discussion.

---

### Official Review · Reviewer_yoip · 2023-10-30

**Soundness:** 3 good
**Presentation:** 3 good
**Contribution:** 3 good
**Rating:** 5
**Confidence:** 5

**Summary:**

The authors claim that they proposed Bongard-OpenWorld, a new benchmark for evaluating real-world few-shot reasoning for machine vision. Based on this benchmark, they further present the few-shot learning baseline approach.

**Strengths:**

The authors claim that they proposed Bongard-OpenWorld, a new benchmark for evaluating real-world few-shot reasoning for machine vision. Based on this benchmark, they further present the few-shot learning baseline approach.

**Weaknesses:**

1. In the experiments, the authors primarily focus on conducting investigations using real-world datasets, particularly the their self-constructed dataset. However, given the Bongard Problem, it raises concerns about the generalizability of the conclusions/findings obtained from real-world datasets to mathematical datasets.

2. The experimental results seems to ignore the traditional models, and it remains a concern.

**Questions:**

Please refer to Weakness.

---

> ### Author Response · Authors · 2023-11-19
> **Responses to Reviewer yoip**
>
> We sincerely thank you for your time and constructive comments. Below, we provide detailed replies to your comments and hope we can resolve your major concerns.
>
> > - In the experiments, the authors primarily focus on conducting investigations using real-world datasets, particularly the their self-constructed dataset. However, given the Bongard Problem, it raises concerns about the generalizability of the conclusions/findings obtained from real-world datasets to mathematical datasets.
>
> Could you please provide a few examples of mathematical datasets? This would allow us to elaborate further. The primary focus of our benchmark is on real-world reasoning, or reasoning in a more realistic setting -- real-world images, open-vocabulary, or open-world concepts. Mathematical datasets, which we suppose primarily adopt synthetic images and close-world concepts, are not the main focus of this work.
>
> > - The experimental results seems to ignore the traditional models, and it remains a concern.
>
> Our experimental results cover a broad spectrum of approaches, ranging from traditional few-shot learning methods, combinations of Large Language Models and Visual Language Models, as well as a neuro-symbol method, and offering a comprehensive analysis. Could you please specify the traditional model you are referring to?  If you give a concrete reference, we would like to try.
>
> We hope the above response can resolve your questions and concerns. Please let us know if there is any further question!

---

> > ### Comment · Reviewer_yoip · 2023-11-22
> > **Official Comment After Rebuttal**
> >
> > Thank you for your reply.
> >
> > As we all know, Bongard-Logo is a good dataset. This study would be more complete if you could add to the results in this aspect.
> >
> > I suggest that the author review the history of meta-learning-based VQA, e.g., [1].
> >
> > [1] Teney, Damien, and Anton van den Hengel. "Visual question answering as a meta learning task." Proceedings of the European Conference on Computer Vision (ECCV). 2018.

---

> ### Comment · Area_Chair_qS2F · 2023-11-22
>
> Hi Reviewer yoip,
>
> Pls read and reply to the authors' responses.
>
> Thanks,
> Cheston

---

### Official Review · Reviewer_bqT5 · 2023-11-01

**Soundness:** 3 good
**Presentation:** 2 fair
**Contribution:** 3 good
**Rating:** 6
**Confidence:** 4

**Summary:**

This paper proposes a new benchmark, called Bongard-OpenWorld that contains visual few-shot reasoning tasks. Specifically, a Bongard problem contains a set of ‘positive’ and ‘negative’ images in the support set, where the positives all share a concept that none of the negatives do. The goal is to use this ‘context’ to infer the positive ‘concept’ in order to correctly label a disjoint set of query images as being positive or negative. While this problem has been studied in previous work, the proposed benchmark differs in that the concepts are ‘open world’ (rather than selected from a predefined small set). Specifically, they leverage Conceptual Concepts which is a massive web-crawled dataset containing image descriptions, and extract concepts from that dataset as well as through crowd-sourcing, to obtain concepts that contain factual or commonsense knowledge. Then, an image search tool is used to find appropriate images from the web to populate the ‘positives’ and ‘negatives’ for each concept (as well as query images) in order to form Bongard problems. They conduct an empirical investigation using both canonical few-shot learning methods as well as leveraging LMs and VLMs in different ways. For example, they explore a scenario where the VLM produces a caption for each of the images in the support set, and then these captions along with the positive and negative labels are fed to the LM which makes a prediction for each query image via in-context learning. This can be done in one-go or iteratively. They also propose a symbolic approach that directly applies logical operations to infer the positive concept.

**Strengths:**

- This paper studies the interesting problem of few-shot visual reasoning
- both ‘traditional’ few-shot learning methods as well as newer ideas involving LMs and VLMs are explored
- the finding that even the best approaches lag significantly behind human performance is an interesting one, and points to the proposed benchmark as a valuable one for pushing the boundaries of of existing methods in this important direction

**Weaknesses:**

- Some related work is missed. [A] (see references below) studies a setting very related to the proposed benchmark (though they didn’t use the terminology Bongard problems). They also created tasks using natural (‘real-world’) images from different datasets, from using computer vision datasets (rather than scraping the web).

- It would be great to add additional few-shot learning baselines to cover families of approaches that are excluded from the current analysis like approaches that perform FiLM-conditioning e.g. [B, C] (see references below) and approaches that train the backbone with gradient descent within each task, like MAML and Proto-MAML (the latter is a version proposed in the Meta-Dataset paper which is cited by this work)

- The paper has some clarity issues, perhaps owing to the fact that the authors tried to ‘squeeze’ a lot of content in the required number of pages. It’s hard to fully understand the different methods by reading only the main paper. I found the neuro-symbolic method proposed especially hard to understand (even after looking at the algorithm in the appendix). Please include some higher-level motivation and the intuition for the particular updates that it entails.

- In Table 2, it’s hard to tell which methods / rows correspond to which of the families of approaches (e.g. a, b, c, or d in Figure 3) – and e.g. which are single-round or multi-round. Perhaps a good way of indicating this is by adding an extra column in that table.

- It would be great to conduct ablation analyses for design choices made in creating the benchmark, like the adversarial query selection that picks the positive query to not be too close to the prototype of the positive class.

- It would be great to conduct an analysis of the effect of the ‘shot’ on these problems. Intuitively, the more positive and negative images the network sees, the easier it is to infer what is the positive class and correctly label query images. Given the negative results in the paper with the current number of shots (6 positives and 6 negatives), in terms of the very large gap from human performance, have the authors considered increasing the number of shots? Understanding how performance of different methods differs as the number of shots increases would be insightful.

- it would also strengthen the paper to tie in the findings of this work with findings in related works. E.g. in the Bongard-HOI benchmark that the authors claim is the most similar, do they have similar findings e.g. in terms of which methods perform better?


Minor
=====
- ‘given 6 positive and 6 negative images [...] (see Figure 1 for illustration)’ – but Figure 1 shows only 3 positive and 3 negative images (6 in total, not each). Maybe clarify that Figure 1 doesn’t correspond to that setting and is used for illustration only? Or describe the task in the intro at a higher level of abstraction, e.g. P positive and N negative images.
- in the caption of Figure 1, highlight ‘hard negatives’ in orange, like ‘distractors’ are highlighted in green, to match the (captions of the) images shown in that figure.
- typo: “prob” → “probe” (on page 6)
- typo: “was not fine-tuning” → “was not fine-tuned” (in Table 2’s caption)

References
=========

- [A] Probing Few-Shot Generalization with Attributes. Ren et al.

- [B] Fast and Flexible Multi-Task Classification Using Conditional Neural Adaptive Processes. Requeima et al. NeurIPS 2019.

- [C] Improved Few-Shot Visual Classification. Bateni et al. 2020

**Questions:**

- In Table 1, how is the number of tasks computed? What constitutes a unique task? Would having the same set of classes but different images in the support set count as the same task?

- In Fig 3a, different few-shot learning algorithms are shown for the classification head only which seemed surprising. Some of these are meta-learning methods that also update the backbone. Is there a meta-training phase (starting possibly from a pretrained architecture) during which the backbone is also finetuned?

- the authors mention that all few-shot learners excluding ChatGPT and GPT-4 use a ConvNext-base. But they also mention that SNAIL uses a transformer architecture. Should SNAIL be listed as another exception there?

- The authors claim that open vocabulary is important for this benchmark and they use this as a  justification for the fact that pretraining on larger datasets leads to better results (“few-shot learners fueled with proper open-ended pretrained models [...] can alleviate this gap”). But an alternative explanation could be that such large pretrained models like CLIP have already seen the specific images and / or concepts presented in the few-shot learning task and thus they simply face a weaker generalization challenge compared to models that were trained on smaller training set which may have a smaller probability of having seen these exact images or concepts. Have the authors made an attempt to examine or rule out this alternative hypothesis?

- Is it possible that some of the created Bongard problems are not solvable? E.g. this could happen if there accidentally is more than one concept that is shared between all of the positive images and none of the negative images. Is care taken to avoid this?

---

> ### Author Response · Authors · 2023-11-19
> **Response to Reviewer bqT5 (1/N)**
>
> We sincerely thank you for your time and constructive comments. Below, we provide detailed replies to your comments and hope we can resolve your major concerns.
>
>
> > - Some related work is missed. [A] (see references below) studies a setting very related to the proposed benchmark (though they didn’t use the terminology Bongard problems). They also created tasks using natural (‘real-world’) images from different datasets, from using computer vision datasets (rather than scraping the web).
>
> We appreciate your suggestion and have incorporated this reference with discussion into our main paper. It is pertinent to note that the primary focus visual concepts of this work comprise various compositional attributes. Our proposed benchmark is characterized by its free-form nature, allowing a wide range of concepts with diverse elements.
>
> > - It would be great to add additional few-shot learning baselines to cover families of approaches that are excluded from the current analysis like approaches that perform FiLM-conditioning e.g. [B, C] (see references below) and approaches that train the backbone with gradient descent within each task, like MAML and Proto-MAML (the latter is a version proposed in the Meta-Dataset paper which is cited by this work).
>
> We are grateful for your suggestions. We are currently implementing the FiLM-conditioning approach, this process will take a while. Regarding the MAML and Proto-MAML approaches, we encountered issues where its training process would collapse, leading to NaN. We believed similar issues have been observed in Bongard-HOI(http://arxiv.org/abs/2205.13803), i.e. some FSL approaches cannot produce meaningful results due to collapsing. Our hypothesis is having even fewer training data (in Bongard-OpenWorld) might have worsen this issue. We hence the exclusion of these results from our report.
>
> > - The paper has some clarity issues, perhaps owing to the fact that the authors tried to 'squeeze' a lot of content in the required number of pages. It’s hard to fully understand the different methods by reading only the main paper. I found the neuro-symbolic method proposed especially hard to understand (even after looking at the algorithm in the appendix). Please include some higher-level motivation and the intuition for the particular updates that it entails.
>
> We have further clarified approaches in the updated paper. For your convinience, here we offer some higher-level motivation and the intuition for the neuro-symbolic method:
>
> - In contrast to canonical FSL and large model based approaches, this approach begins with not just converting images into captions (ex. a dog is running on the grass), but further into concepts (ex. <dog, running, on grass>). This is done by employing GPT-4, the prompt is detailed in Appendix D. Our goal here is to mitigate the substantial noise present in image captions (both in single-round and multi-round approaches).
> - With these concepts, we apply logical operations like AND, OR, and NOT, aiming at mimicking the human logical thinking process when tacking Bongard problems. Specifically (please note that this is just high-level illustration and some details for handling corner cases are omitted. Again, the [code](https://github.com/Bongard-OpenWorld/Bongard-OpenWorld) includes all the implementation details):
>     1. Compute the intersection of the concepts of all positive images, this gives us the initial hypothesis on the true concept of this Bongard problem. If there is no intersection (possibly due to flaw in perception), a majority vote is adopted -- concepts that present in more than 4 positive images will be treated as the "intersection".
>     2. For each negative image, we substract its concept from the "intersection" obtained in the previous step, this is to mimick how human rule out the concepts that is depicted by both the positives and the negatives, and finally reach to the concept that is uniquely depicted by the positives.
>     3. We compare the updated "intersection" with the concept of the query image. If the "intersection" is a subset of the concepts depicted by the query, the query will be categorized as positive, otherwise deemed negative.
>     4. The "intersection" is also the ultimate induced concept of this Bongard problem, exclusively depicted by the positives.
>
> - As a result, such logical operations over concepts can deterministically produce binary predictions on the query image and induced visual concepts depicted by the positives.
>
> Negatives contribute candidates that aid positives in updating their missed atomic concepts (potentially distractors). Conversely, positives provide candidates that help negatives refine their concepts (potentially overlaps).

---

> > ### Author Response · Authors · 2023-11-19
> > **Response to Reviewer bqT5 (2/N)**
> >
> > > - In Table 2, it’s hard to tell which methods / rows correspond to which of the families of approaches (e.g. a, b, c, or d in Figure 3) – and e.g. which are single-round or multi-round. Perhaps a good way of indicating this is by adding an extra column in that table.
> >
> > We appreciate your insightful suggestion and have updated Table 2.
> >
> > > - It would be great to conduct ablation analyses for design choices made in creating the benchmark, like the adversarial query selection that picks the positive query to not be too close to the prototype of the positive class.
> >
> > We have added an illustration (see Figure 11 in Appendix G) to visually demonstrate the ablation analysis of design choices. Without the adversarial query selection, and relying solely on randomization, the query images become easily distinguishable (all images and concepts are not changed).
> >
> > For instance, in subfigure (a), the positive query and positive images are not similar in appearance, only sharing the color of "red". However, negative images also feature "red" and "tie" (partially overlap), but none depict "red tie" (exclusively depicted by the positives), necessitating complex reasoning for correct determination. If only by appearance, the negative query could be easily misclassified as positive due to its resemblance to P.1, P.3, and P.4 (from left to right), and even P.2, resulting in it is very difficult to disentangle the problem.
> >
> > Conversely, in subfigure (b), without adversarial query selection, simplifies this matter significantly. The positive query in (b) closely resembles P.1, P.2, P.3, and even P.4, facilitating a straightforward determination. Similarly, the negative query in (b) bears a high resemblance to N.1, N.2, and N.3 in appearance.
> >
> >
> > > - It would be great to conduct an analysis of the effect of the ‘shot’ on these problems. Intuitively, the more positive and negative images the network sees, the easier it is to infer what is the positive class and correctly label query images. Given the negative results in the paper with the current number of shots (6 positives and 6 negatives), in terms of the very large gap from human performance, have the authors considered increasing the number of shots? Understanding how performance of different methods differs as the number of shots increases would be insightful.
> >
> > We have updated the results to include 2, 3, 4, and 5-shot in support set on SNAIL, in contrast to the initially reported 6-shot approach. All of them employ identical settings and parameters, utilizing OpenCLIP pretrained image representation along with a training caption head. It indeed aligns with intuition. One thing to note: although reducing the number of shots could lead to more significant challenge, but it also deviate from the original setting of the Bongard problems, which we respect in Bongard-OpenWorld.
> >
> > | x-shot | short concept | long concept | CS* concept | non-CS* concept | avg. |
> > | :---: | :---: | :---: | :---: | :---: | :---: |
> > | 2 | 60.6 | 56.6 | 52.7 | 61.0 | 58.8 |
> > | 3 | 62.4 | 58.8 | 57.3 | 62.1 | 60.8 |
> > | 4 | 61.9 | 61.6 | 62.7 | 61.4 | 61.8 |
> > | 5 | 64.7 | 63.2 | 62.7 | 64.5 | 64.0 |
> > | 6 | 66.1 | 61.5 | 63.6 | 64.1 | 64.0 |

---

> > > ### Author Response · Authors · 2023-11-19
> > > **Response to Reviewer bqT5 (3/N)**
> > >
> > > > - it would also strengthen the paper to tie in the findings of this work with findings in related works. E.g. in the Bongard-HOI benchmark that the authors claim is the most similar, do they have similar findings e.g. in terms of which methods perform better?
> > >
> > > We thank you for your suggestion! We've added discussions to the Appendix E (under "Results compared to relavent benchmark, ex. Bongard-HOI"). For your convinience, here are some highlights: Compared to Bongard-HOI, there are both shared observations and differences.
> > >
> > > 1) shared observations: canonical FSL (ex. SNAIL, MetaBaseine) do not work well on both datasets, regardless what representation backbone is being used.
> > >
> > > 2) differences:
> > >
> > >     - the most effective methods differ (Bongard-OpenWorld is SNAIL vs. Bongard-HOI is Meta-Baseline);
> > >     - Bongard-HOI is more sensitive to open-vocabulary pretrained representations (its concept vocabulary is limited); Bongard-OpenWorld is not as it is designed to be open vocabulary or open-world;
> > >     - Bongard-HOI does not have results of LLM-based methods.
> > >
> > >     The first and last observations are obvious. Here we would like to elaborate a bit on the second one. We offered some additional results of benchmarking CLIP (shown below), a relatively more free-form and open-world visual backbone on both Bongard-HOI (a prior work, focus on compositional generalization on closed-world concepts) and Bongard-Openworld. As you can see, with a much stronger backbone like OpenCLIP, ProtoNet can achieve 71.3% acc on Bongard-HOI (way higher than the previous Bongard-HOI SOTA 58.3% using ImageNet-1k pretrained backbone), but on Bongard-OpenWorld it can only reach 58.3%. In fact, Bongard-HOI discourages the use of such an open-world pretrained backbone like CLIP because it might undermine the challenges posed by the benchmark (compositional generalization will not be an issue anymore, as all concepts in training/generalization test have been saw by CLIP). Rather, Bongard-OpenWorld allows and indeed encourages all types of backbone and models by introducing free-form open-world concepts. And as the additional result suggest, we're successful -- even the powerful CLIP backbone is still far from beating our benchmark.
> > >
> > >     | benchmark | method | image representation | aux. task? | avg. |
> > >     | :---: | :---: | :---: | :---: | :---: |
> > >     | Bongard-OpenWorld | ProtoNet | ImageNet-1k | &#10007; | 56.0 |
> > >     | Bongard-OpenWorld | ProtoNet | OpenCLIP | &#10007; | 58.3 |
> > >     | Bongard-HOI | ProtoNet | ImageNet-1k | &#10007; | 58.3 |
> > >     | Bongard-HOI | ProtoNet | OpenCLIP | &#10007; | 71.3 |
> > >
> > > > - 'given 6 positive and 6 negative images [...] (see Figure 1 for illustration)' – but Figure 1 shows only 3 positive and 3 negative images (6 in total, not each). Maybe clarify that Figure 1 doesn’t correspond to that setting and is used for illustration only? Or describe the task in the intro at a higher level of abstraction, e.g. P positive and N negative images.
> > > > - in the caption of Figure 1, highlight ‘hard negatives’ in orange, like ‘distractors’ are highlighted in green, to match the (captions of the) images shown in that figure.
> > > > - typo: “prob” → “probe” (on page 6)
> > > > - typo: “was not fine-tuning” → “was not fine-tuned” (in Table 2’s caption)
> > >
> > > Thanks for your meticulous observation. We have clarified it in our main paper, Figure 1 is intended exclusively to depict the Bongard-OpenWorld Problems for a better view. Actually, each Bongard-OpenWorld Problem comprises six positive and six negative images, along with a corresponding query image separately. Highlight and typos have all been corrected.
> > >
> > > > - In Table 1, how is the number of tasks computed? What constitutes a unique task? Would having the same set of classes but different images in the support set count as the same task?
> > >
> > > In Table 1, each dataset sample is a task. For Bongard-OpenWorld, a task is characterized as a Bongard Problem, which includes a support set of six positive and six negative images along with two query images separately (one positive, one negative), a total of (6+6+2) images with the common visual concept is exclusively depicted by images from the positive set. Please note that when computing accuracy, we treat one Bongard problem as two test cases, one with the positive query and one with the negative query.
> > > Compared to CC-3M, where each image-text pair constitutes a task that contains numerous recurring concepts, in Bongard-OpenWorld, we ensure that each task encompasses a **unique** common visual concept across the whole dataset, thereby the situation in your last question does not exist.

---

> > > > ### Author Response · Authors · 2023-11-19
> > > > **Response to Reviewer bqT5 (4/N)**
> > > >
> > > > > - In Fig 3a, different few-shot learning algorithms are shown for the classification head only which seemed surprising. Some of these are meta-learning methods that also update the backbone. Is there a meta-training phase (starting possibly from a pretrained architecture) during which the backbone is also finetuned?
> > > >
> > > > We apologize for the confusion caused by Figure 3a, it has been updated for clarity, a flame icon indicates trainable components, while a snowflake icon denotes those that are frozen. In reality, our training process also involves updating weights of the backbone. Detailed information about the learning rate is available in the Appendix D. In our abstract, we have clearly stated that all the code and the dataset have been made public under this anonymous github repo: https://github.com/Bongard-OpenWorld/Bongard-OpenWorld. Feel free to check.
> > > >
> > > > > - the authors mention that all few-shot learners excluding ChatGPT and GPT-4 use a ConvNext-base. But they also mention that SNAIL uses a transformer architecture. Should SNAIL be listed as another exception there?
> > > >
> > > > The image encoder of the SNAIL method mentioned in our paper is ConvNext-base, and its reasoning component is indeed using a transformer architecture.
> > > >
> > > > > - The authors claim that open vocabulary is important for this benchmark and they use this as a justification for the fact that pretraining on larger datasets leads to better results (“few-shot learners fueled with proper open-ended pretrained models [...] can alleviate this gap”). But an alternative explanation could be that such large pretrained models like CLIP have already seen the specific images and / or concepts presented in the few-shot learning task and thus they simply face a weaker generalization challenge compared to models that were trained on smaller training set which may have a smaller probability of having seen these exact images or concepts. Have the authors made an attempt to examine or rule out this alternative hypothesis?
> > > >
> > > > Thanks for your insight and we're sorry for not making this clear. Indeed, it is difficult entirely rule out the alternative explanation due to the extensive scale of the CLIP training dataset (if we're talking about the OpenAI CLIP, it is impossible as its training dataset remained undisclosed). But we would like to point out that one goal of Bongard-OpenWorld is to evaluate this context-dependent reasoning capability (as mentioned in Bongard-HOI as well) in a more **realistic** setting, which should include a much larger or even unlimited size of concept vocabulary than canonical close-world settings as in the original Bongard problem or Bongard-HOI. Therefore, we believe that no matter if the models indeed exhibit better compositional generalization on understanding visual concepts (which is more ideal) or they perform better possibly due to being exposed to some images/concepts during training, the observation that open-ended pretrained models genearlly perform better in this more **realistic** Bongard setting as in Bongard-OpenWorld should not be affected.
> > > >
> > > >
> > > > > - Is it possible that some of the created Bongard problems are not solvable? E.g. this could happen if there accidentally is more than one concept that is shared between all of the positive images and none of the negative images. Is care taken to avoid this?
> > > >
> > > > Each Bongard Problem in our proposed benchmark has undergone extensive manual review to ensure full compliance with task settings. We have exercised due diligence to avoid including any unsolvable problems. Thanks for your attention to this matter.
> > > >
> > > > We hope the above response can resolve your questions and concerns. Please let us know if there is any further question!

---

> > > > > ### Comment · Reviewer_bqT5 · 2023-11-22
> > > > > **response to authors**
> > > > >
> > > > > Hi authors,
> > > > >
> > > > > Thank you for the very detailed response!
> > > > >
> > > > > I appreciate the discussion of other families of approaches, clarifications, comparison of the findings to those on Bongard-HOI, and discussions on the issue that some models may have been trained on the same concepts that appear in bongard tasks which may act as a confounding factor for some conclusions. I appreciate that the benchmark aims for realism and practicality, but i think in light of this confounding factor, we can't make claims like the open-vocabulary nature being responsible for better results on this benchmark (without ruling out the alternative hypothesis that those models had actually seen the same concepts so they simply face an easier generalization problem). So adjusting claims to acknowledge this is important.
> > > > >
> > > > > Overall, the authors have answered several of my questions and i maintain my opinion of weak acceptance. The reason I don't further increase my score is that I think the paper could further improve in terms of clarity and presentation, and i also agree with concerns brought up by Reviewer CSkd on strengthening the motivation of the particular design choices made in creating this benchmark and understanding the limitations (e.g. does it bias towards one family of models).

---

> > > > > > ### Author Response · Authors · 2023-11-23
> > > > > >
> > > > > > Dear reviewer,
> > > > > >
> > > > > > We sincerely appreciate your insightful suggestions and acknowledgement of our rebuttal. We are currently engaging in further discussions with Reviewer CSkd. Your continued interest and input in this process would be greatly appreciated.
> > > > > >
> > > > > > Best, Authors

---

> ### Comment · Area_Chair_qS2F · 2023-11-22
>
> Hi Reviewer bqT5,
>
> Pls read and reply to the authors' responses.
>
> Thanks,
> Cheston

---

> > ### Author Response · Authors · 2023-11-23
> >
> > Dear AC,
> >
> > Thank you for your time, your consideration, and all the support you offer on our submission during the review and discussion period. We're grateful to have you supervise the reviewing process of our submission and the reviewer's comments as well.
> >
> > We believe it is your help that ultimately drives the reviewers to response to our initial rebuttal. They are very encouraging and constructive, we have tried our best to promptly respond to them, with additional analysis, clarifications, and results. We are glad to see many of their concerns have been addressed.
> >
> > We are currently continuing our discussions with the reviewers and deeply appreciate your continued involvement in this process.
> >
> > Sincerely,
> > Authors

---

### Author Response · Authors · 2023-11-19
**General Response to all Reviewers**

We thank all reviewers for their insightful comments and acknowledgment of our contributions. We highlight the major contributions of our work as follows:

- We propose "a new benchmark for few-shot visual reasoning" (rephrasd from Reviewer yoip), it "tackles an important shortcoming of current methods to understand fine-grained semantic concepts in contrast to hard negatives" (Reviewer CSkd). Our benchmark improves from the prior Bongard-HOI work by considering (1) free-form and (2) "open world" (rather than selected from a predefined small set) (Reviewer bqT5). "It promotes research into the few-shot reasoning capabilities of current black box deep learning models" (Reviewer Gtsv).
- Our new benchmark "serves as a litmus test for current limitations in visual intelligence, motivating further research toward enhancing few-shot reasoning in visual agents" (Reviewer Gtsv), and "even the best approaches lag significantly behind human performance" (Reviewer bqT5), which indeed "makes it a challenging setting for new methods to be developed in the future" (Reviewer CSkd).
- We conduct benchmarking with "a comprehensive assessment" (Reviewer Gtsv), where "both 'traditional' few-shot learning methods as well as newer ideas involving LMs and VLMs are explored" (Reviewer bqT5). Specifically, it "includes the evaluation of four different kinds of approaches that include a few shot learning approaches, LLM+VLM in a single step, LLM+VLM in multiple iteration steps, and finally a proposed neuro-symbolic architecture" (Reviewer Gtsv) and indeed "showing that the best models (64%) are still far from human performance (91%)" (Reviewer ZDnE).

We've revised our manuscript per the reviewers' suggestions (highlighted in red in the uploaded revision pdf). Detailed responses to each reviewer's concerns are carefully addressed point-by-point. Below summarize the major updates we've made:

- Paper citations. (**references**)
	> - [A] Probing Few-Shot Generalization with Attributes. Ren et al.
	> - [B] Fast and Flexible Multi-Task Classification Using Conditional Neural Adaptive Processes. Requeima et al. NeurIPS 2019.
	> - [C] Improved Few-Shot Visual Classification. Bateni et al. 2020.

- Clarification of Bongard-OpenWorld problem setting. (**Figure 1**)

- Explanation for the statistics of Bongard-OpenWorld. (**Figure 2**)

- Clarification of the process for dataset collection from the CC-3M. (**section 2.1**)

- Clarification of main methods. (**Figure 3 (a), (b)**)

- Broader discussion and explanation for highlighting Bongard-OpenWorld's challenges. (**Appendix B**)

- References of caption metrics and exact instructions to annotators. (**Appendix C**)

- All details of prompts, including the latest powerful GPT-4V. (**Appendix D**)

- Compared results to Bongard-HOI and additional discussion. (**Appendix E.2**)

- Results of imbalance between positive and negative queries (**Appendix E.3**) & analysis of natural baseline. (**Appendix E.4**)



- Parameter details of the few-shot learning models. (**Table 6**)

- More comprehensive qualitative results and failure cases (**Appendix G**), including an ablation analysis of adversarial query selection. (**Figure 11**)

We believe our benchmark could make a timely contribution to the visual reasoning & understanding community and would like to be involved in further discussions if any question is raised.

Best,
Authors.

---

### Meta-Review · Area_Chair_qS2F · 2023-12-06

**Metareview:**

This was a very interesting piece of work, which reviewers found to be robust and very comprehensive, particularly with the evaluation framework covering four different settings/approaches and many recent models. Designed specifically to be challenging via adversarial selection of examples, the best models are still very far from human performance.

This AC particularly finds the chosen variation of the Bongard paradigm to be illuminating. There was some discussion over the design choices and their justification, but taking the broader view, my opinion is that as long as humans can perform the task (they do get 91% accuracy) and the input/output setting of the task are comparable for humans and models, it's a valid benchmark.

What's interesting is how the task uses a very simple setup to force the model or human to not only induce open-ended concepts from few examples, but the negative set can be used to eliminate simple first-order concepts as viable answers. At least for humans, it is likely that they need to maintain multiple hypotheses or "backtrack" in the search space when a seemingly viable answer is negated. Possibly this is why existing models do not do well, suggesting an important human cognitive capability that should be added to future models.

In terms of weaknesses, reviewers initially had valid concerns or suggestions about addtional models (e.g. GPT4V), the imbalanced problem setting (and more generally the design choices behind this variation of the Bongard task), missing simple baselines and the relatively small dataset. The authors responded comprehensively with additional results and analyses, detailed explanations and significant revisions to the paper. Ultimately, the majority of reviewers found their major concerns to be addressed, as reflected both quantitatively and qualitatively.

Overall, this was a solid benchmarking paper, which I personally find to be very interesting. I believe it has potential for lasting impact to the broader community, and it deserves acceptance to ICLR.

**Justification For Why Not Higher Score:**

There were some remaining minor concerns about readability, as well as clearer explanation of design choices.

**Justification For Why Not Lower Score:**

Very interesting and novel work, with potential for lasting significance.

---

### Decision · Program_Chairs · 2024-01-16

Accept (poster)